# Phagocytic 'teeth' and myosin-II 'jaw' power target constriction during phagocytosis

**Daan Vorselen[1†], Sarah R Barger[2†‡], Yifan Wang[3], Wei Cai[3], Julie A Theriot[1]\*, Nils C Gauthier[4], Mira Krendel[2]\***

[1]Department of Biology and Howard Hughes Medical Institute, University of Washington, Seattle, United States; [2]Department of Cell and Developmental Biology, State University of New York Upstate Medical University, Syracuse, United States; [3]Department of Mechanical Engineering, Stanford University, Stanford, United States; [4]IFOM, FIRC Institute of Molecular Oncology, Milan, Italy

**Abstract** Phagocytosis requires rapid actin reorganization and spatially controlled force generation to ingest targets ranging from pathogens to apoptotic cells. How actomyosin activity directs membrane extensions to engulf such diverse targets remains unclear. Here, we combine lattice light-sheet microscopy (LLSM) with microparticle traction force microscopy (MP-TFM) to quantify actin dynamics and subcellular forces during macrophage phagocytosis. We show that spatially localized forces leading to target constriction are prominent during phagocytosis of antibody-opsonized targets. This constriction is largely driven by Arp2/3-mediated assembly of discrete actin protrusions containing myosin 1e and 1 f ('teeth') that appear to be interconnected in a ring-like organization. Contractile myosin-II activity contributes to late-stage phagocytic force generation and progression, supporting a specific role in phagocytic cup closure. Observations of partial target eating attempts and sudden target release via a popping mechanism suggest that constriction may be critical for resolving complex in vivo target encounters. Overall, our findings present a phagocytic cup shaping mechanism that is distinct from cytoskeletal remodeling in 2D cell motility and may contribute to mechanosensing and phagocytic plasticity.

**\*For correspondence:**
jtheriot@uw.edu (JAT);
krendelm@upstate.edu (MK)

[†]These authors contributed equally to this work

**Present address:** [‡]Molecular, Cellular, Developmental Biology, Yale University, New Haven, United States

**Competing interest:** The authors declare that no competing interests exist.

## Editor's evaluation

This study provides interesting new insights into the roles of mechanical forces generated by actin polymerization and myosin-II contractility in Fc receptor-mediated phagocytosis. The authors identify F-actin 'teeth', which are important for microparticle constriction throughout the phagocytosis process. Moreover, they elucidate the specific roles of Arp2/3 nucleated actin networks, and myosin-II-based contractile structures in phagocytosis.

## Introduction

Phagocytic uptake of microbial pathogens, apoptotic cells, and debris are essential processes for human health (*Boada-Romero et al., 2020*; *Lim et al., 2017*). Given the variety of phagocytic targets, ranging widely in shape, size, and mechanical stiffness, this process requires remarkable plasticity. Phagocytosis is initiated when phagocytic receptors recognize distinct molecular patterns coating the target (*Freeman and Grinstein, 2014*; *Uribe-Querol and Rosales, 2020*). Selective engagement and specific exclusion of phagocyte receptors (*Freeman et al., 2018*; *Freeman et al., 2016*) enables downstream signaling to initiate the formation of membrane protrusions, which are guided around

the target through sequential ligand engagement in a zipper-like fashion to form the phagocytic cup (*Griffin and Silverstein, 1974*; *Jaumouillé and Waterman, 2020*; *Swanson and Hoppe, 2004*). Mechanically, progression of the phagocytic cup is powered by F-actin polymerization that pushes the plasma membrane forward and culminates in the eventual closure of the cup and the formation of a membrane-enclosed phagosome (*Vorselen et al., 2020a*).

Previously, F-actin within the phagocytic cup was generally considered to be homogenous and the membrane extensions around the target were frequently likened to lamellipodia at the leading edge of a migrating cell (*Blanchoin et al., 2014*; *Davidson and Wood, 2020*; *Jaumouillé and Waterman, 2020*; *Small et al., 2002*). Yet recent studies have identified dynamic adhesions or podosome-like structures within the phagocytic cup that appear to be sites of actin polymerization (*Barger et al., 2019*; *Ostrowski et al., 2019*). This suggests a fundamentally different mechanism for cup shaping based on assembly of individual actin-based protrusions versus a uniform actin meshwork. Podosomes are specialized F-actin adhesive structures that are prominent in myeloid cells and capable of generating forces and degrading extracellular matrix (*Linder, 2007*; *van den Dries et al., 2019a*). Interestingly, podosomes are mechanosensitive (*Labernadie et al., 2014*; *van den Dries et al., 2019b*), generating greater forces in response to stiffer substrates, and thus podosome-like structures may contribute to the mechanosensitivity observed in phagocytosis, whereby phagocytes engulf stiffer targets more readily than softer ones (*Beningo and Wang, 2002*; *Jaumouillé et al., 2019*; *Sosale et al., 2015*; *Vorselen et al., 2020b*; *Vorselen et al., 2020a*). The 'core' structure of podosomes is primarily composed of branched F-actin nucleated by the Arp2/3 complex. The Arp2/3 complex itself was initially considered to be critical for all modes of phagocytosis (*May et al., 2000*), although subsequent work using $Arpc2^{-/-}$ macrophages has revealed that in Fc receptor (FcR)-mediated phagocytosis, a dendritic actin network is only critical for the ingestion of larger targets (>6 μm) (*Rotty et al., 2017*). This inhibition of phagocytosis upon loss of the Arp2/3 complex activity may relate to the cell's ability to form IgG-FcR microclusters at the earliest stages of FcR signaling (*Jo et al., 2021*).

The mechanism of phagocytic cup closure and the involvement of specific myosin motor proteins therein has also remained elusive (*Barger et al., 2020*; *Jaumouillé and Waterman, 2020*). The distinct localization of myosin-I at the rim of the phagocytic cup has led to a speculated role in cup closure (*Barger et al., 2019*; *Swanson et al., 1999*; *Ikeda et al., 2017*). Myosin-II has also been hypothesized to orchestrate a 'purse-string' contractility based on observations of phagocytes deforming red blood cells during internalization (*Araki et al., 2003*; *Barger et al., 2020*; *Swanson et al., 1999* ). However, the effect of myosin-II inhibition on phagocytic uptake efficiency is highly variable, with some studies showing no effect (*Jaumouillé et al., 2019*; *Rotty et al., 2017*) and others reporting a 40–80% reduction (*Olazabal et al., 2002*; *Sosale et al., 2015*; *Tsai and Discher, 2008*; *Yamauchi et al., 2012*). Myosin-II inhibition also has limited effect on traction forces tangential to the target surface in 2D spreading assays known as frustrated phagocytosis (*Kovari et al., 2016*). Together these studies show that the actin dynamics within the membrane extensions of the phagocytic cup must be finely tuned for successful engulfment, but the role of specific myosin motor proteins and any associated contractility in cup closure remains to be elucidated.

It is clear that phagocytosis is driven by mechanical forces, but examining these forces has been challenging due to the limitations of experimental approaches. Changes in cellular tension during phagocytosis measured by micropipette aspiration have been well studied (*Heinrich, 2015*; *Herant et al., 2005*), but this technique fails to capture cell-target forces and the spatial variation within the phagocytic cup. Recently, traction force microscopy (TFM) combined with frustrated phagocytosis assays has been used to measure cell-target forces (*Barger et al., 2019*; *Jaumouillé et al., 2019*; *Kovari et al., 2016*; *Rougerie and Cox, 2020*), however, this assay fails to capture the biologically relevant geometry of the phagocytic cup, which likely affects cytoskeletal dynamics and force generation. Moreover, in its most common application, TFM only measures forces tangential to the target surface, neglecting forces normal to the target surface, which may be critical. With the recent introduction of particle-based force sensing methods (*Mohagheghian et al., 2018*; *Träber et al., 2019*), particularly microparticle traction force microscopy (MP-TFM) (*Vorselen et al., 2020b*), both normal and shear force components can now be studied throughout phagocytosis.

Here, we utilize live-cell imaging combining MP-TFM and lattice light-sheet microscopy (LLSM) to reveal how mechanical forces generated by actin polymerization and myosin contractility drive phagocytic engulfment mediated by FcR. We show that phagocytes assemble F-actin 'teeth' that

mediate target constriction throughout phagocytosis. Analysis of forces shows a unique signature, in which target constriction, or squeezing, is balanced by pulling forces at the base of the phagocytic cup at early stages and target compression throughout the cup at later stages. Together, normal forces far exceed shear forces at the cell-target interface, pointing to a mechanism fundamentally distinct from lamellipodial spreading in cell motility. We find that target constriction is mediated by Arp2/3-mediated actin polymerization throughout phagocytosis. Moreover, based on both force analysis and precise quantitative measurement of cup progression, we establish a clear role for myosin-II purse string contractility, specifically in phagocytic cup closure. Finally, we present how this force signature might be critical for target selection and ingestion in more complex physiological settings.

## Results

### LLSM reveals sequence of target deformations induced during live phagocytic engulfment

Given the fast, 3D, and light-sensitive nature of phagocytosis, we used LLSM for high-speed volumetric imaging with minimal phototoxicity to investigate cytoskeletal dynamics and phagocytic forces. To monitor internalization in real time, RAW264.7 macrophages were transfected with mEmerald-Lifeact for labeling of filamentous actin and were fed deformable acrylamide-*co*-acrylic acid-microparticles (DAAMPs) (*Figure 1a*). To investigate FcR-mediated phagocytosis, DAAMPs were functionalized with BSA and anti-BSA IgG, as well as AF647-Cadaverine for visualization (*Vorselen et al., 2020b*; *Figure 1b*). RAW macrophages were fed DAAMPs with a diameter of 9 μm and a Young's modulus of 1.4 or 6.5 kPa, which is in the same range as healthy tissue cells and physiological targets of macrophages like apoptotic cells (*Cross et al., 2007*; *Van der Meeren et al., 2020*). RAW macrophages typically formed a chalice-shaped phagocytic cup and completed phagocytosis in a similar timeframe (~3 min) as previously reported for much stiffer (2–3 GPa) polystyrene particles (*Horsthemke et al., 2017*; *Figure 1a*, *Video 1*, *Figure 1—figure supplement 1*). 3D shape reconstructions of the DAAMPs enabled us to examine target deformations as a direct readout of phagocytic forces in real time (*Figure 1b–c*). We observed target constriction defined by discrete spots of deformation that appeared around the circumference of the DAAMP at the rim of the phagocytic cup (*Figure 1d*, *Video 2*). While these deformations were more apparent using the softer 1.4 kPa DAAMPs, qualitatively similar force patterns were observed using stiffer 6.5 kPa targets (*Figure 1—figure supplement 2*, *Video 3*) and previously using 0.3 kPa targets with fixed J774 macrophages (*Vorselen et al., 2020b*). Interestingly, these indentations traveled parallel to the direction of phagocytic cup elongation, along the length of the DAAM particle until cup closure and were associated with ~400 nm maximum target constriction for 1.4 kPa DAAMPs (*Video 2*, *Figure 1—figure supplement 1*). To determine whether this behavior was also present in primary macrophages, we also imaged bone marrow-derived macrophages (BMDMs) transfected with mEmerald-Lifeact and observed a similar mechanical progression, albeit with smaller deformations (*Figure 1—figure supplement 3*). In addition to local deformations in RAW macrophages, we also observed bulk compressive stresses during phagocytic cup progression (~0.5 kPa) and after complete internalization of the DAAMPs (up to 1.5 kPa), leading to a dramatic reduction in DAAMP diameter (*Figure 1—figure supplement 1*). The spherical appearance of targets and the gradual monotonic increase in compression after completion of engulfment suggests that this compression may relate to the recently observed shrinkage of internalized macropinosomes by osmotic pressure changes regulated by ion flux (*Freeman et al., 2020*). However, we sometimes observed the appearance of an F-actin shell around the target, similar to previous reports (*Liebl and Griffiths, 2009*), suggesting that cytoskeletal forces may also contribute to such compression (*Figure 1—figure supplement 1*, *Video 4*).

### Forces normal to the target surface are dominant during phagocytic engulfment and lead to strong target constriction

To more closely investigate the role of the actin cytoskeleton in force generation during FcR-mediated engulfment, we performed MP-TFM measurements on fixed cells during phagocytosis. RAW macrophages were exposed to 9 μm, 1.4 kPa DAAMPs, after which they were fixed and stained for F-actin (*Figure 2a*). Immunostaining of the exposed particle surface allowed precise determination of the stage of engulfment (*Figure 2—figure supplement 1*). Confocal z-stacks of phagocytic cups enabled

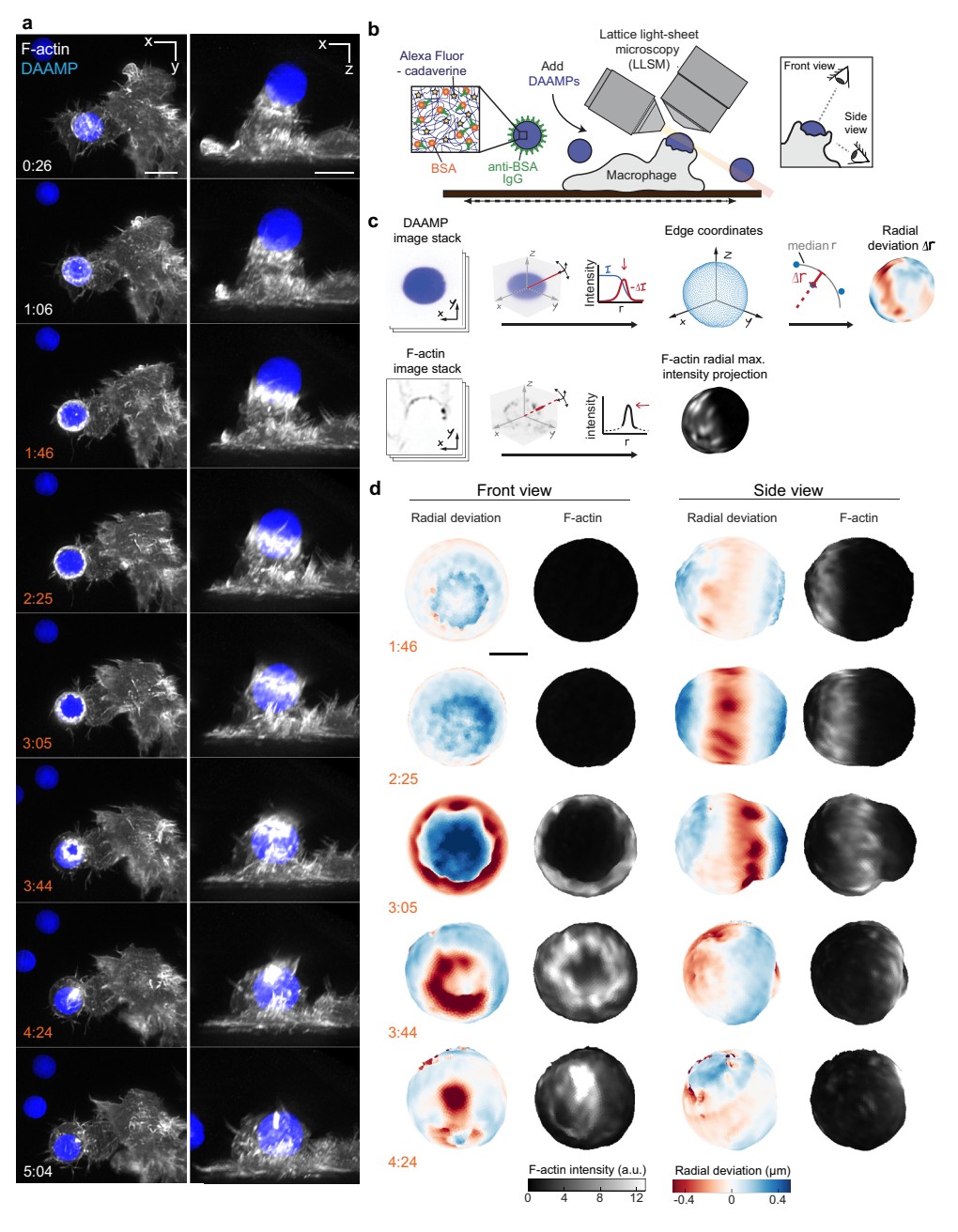

**Figure 1.** Combined microparticle traction force microscopy (MP-TFM) and lattice light-sheet microscopy (LLSM) reveals phagocytic force-induced deformations in real time. RAW macrophages transfected with mEmerald-Lifeact were fed soft deformable acrylamide-*co*-acrylic acid micro (DAAM)-particles (9 μm,1.4 kPa) functionalized with IgG and AF647-Cadaverineand imaged using LLSM. (**a**) Time lapse montage (min:s) of maximum intensity projections in x/y and x/z. Scale bar, 5 μm. (**b,c**) Schematic of the combined LLSM and MP-TFM experimental approach and analysis, respectively. (**d**) Front and side view of reconstructed DAAM-particle internalized in (**a**) showing target deformations and F-actin localization on particle surface. Colorscale represents the deviation of each vertex from a perfect sphere with radius equal to the median radial distance of edge coordinates to the particle centroid. Scale bar, 3 μm.

The online version of this article includes the following source data and figure supplement(s) for figure 1:

**Source data 1.** Numeric data for *Figure 1—figure supplement 1a–c,e,f* and *Figure 1—figure supplement 2c–f*.

**Figure supplement 1.** Constriction and bulk compressive stresses during phagosome formation and maturation.

*Figure 1 continued on next page*

*Figure 1 continued*

**Figure supplement 2.** Phagocytosis of stiffer targets follows a qualitatively similar mechanical progression as for softer targets.

**Figure supplement 3.** Murine bone marrow-derived macrophage (BMDM) phagocytosis is mechanically qualitatively similar to RAW macrophage phagocytosis.

3D target shape reconstructions with super-resolution accuracy and inference of cellular traction forces (*Figure 2b and c*). The deformation patterns, although slightly lower in magnitude for the fixed samples (*Figure 1—figure supplement 1*), were similar to those observed in living cells by the LLSM imaging (*Figure 2b* and *Figure 1—figure supplement 1*). Specifically, we noted a ring of inhomogeneous F-actin localized along the rim of the phagocytic cup, where high F-actin intensity strongly correlated with inward target deformation.

Force analysis (see Materials and methods) revealed compressive stresses up to 400 Pa at these sites (*Figure 2c*), which is substantially greater than our previous findings of stresses of ~100 Pa using softer targets with Young's modulus 0.3 kPa (*Vorselen et al., 2020b*), and suggests that force exertion during phagocytosis may be regulated based on target rigidity. The total compressive forces in the phagocytic rim, which lead to target constriction, increased from ~1 nN in early-stage phagocytosis (fraction engulfed ~22%) to ~10 nN in later stages (fraction engulfed >50%). The shear forces were consistent in magnitude with reported values using TFM during frustrated phagocytosis on planar gels of similar rigidity (*Rougerie and Cox, 2020*) and were ~7 -fold lower than the observed normal forces, independent of the stage of phagocytic engulfment (*Figure 2d*). This suggests that normal forces dominate the mechanical interaction in phagocytosis, which is in stark contrast with lamellipodial extensions during cell migration where shear forces dominate (*Case and Waterman, 2015*; *Legant et al., 2013*).

## Target constriction coincides with sites of F-actin accumulation and increases with uptake progression

To identify overall trends in F-actin distribution and location of target deformations within the phagocytic cup, we aligned the 3D images of phagocytic cups and analyzed profiles along the phagocytic axis, defined as the axis from the centroid of the cell-target contact area through the target centroid to the opposing target surface (*Figure 2e*). This analysis confirmed a clear accumulation of F-actin near the front of the phagocytic cup (~5 -fold higher than at the cup base), which precisely colocalized with the site of maximal applied inward normal forces regardless of engulfment stage, as illustrated by quantifying the surface mean curvature (*Figure 2f*). A similar distribution of F-actin and deformation was observed with stiffer 6.5 kPa targets, albeit with smaller target deformations, and

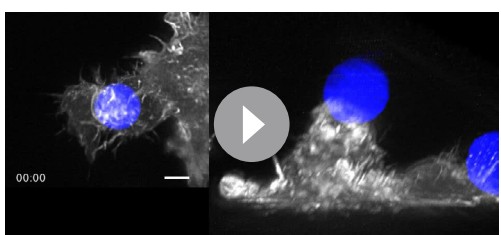

**Video 1.** RAW macrophage ingesting deformable acrylamide-*co*-acrylic acid-microparticle (DAAMP) imaged by lattice light-sheet microscopy. RAW macrophages transfected with mEmerald-Lifeact (gray) were fed DAAM-particles (9 μm,1.4 kPa) (blue) functionalized with AF647-Cadaverine, BSA, and anti-BSA IgG. Maximum intensity projections in xy (left) and xz (right). Lower left time stamp: min:s. Scale bar, 5 μm.
https://elifesciences.org/articles/68627/figures#video1

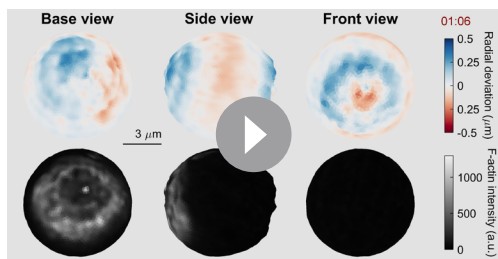

**Video 2.** 3D microparticle shape reconstruction shows phagocytic force-induced deformations in real time. Base, side, and front views of reconstructed deformable acrylamide-*co*-acrylic acid-microparticle (DAAMP) internalized in Figure 1, Video 1 showing target deformations (above) and F-actin localization on particle surface (below). Color scales for radial deviation and F-actin intensity shown on right. Upper right time stamp: min:s. Scale bar, 3 μm.
https://elifesciences.org/articles/68627/figures#video2

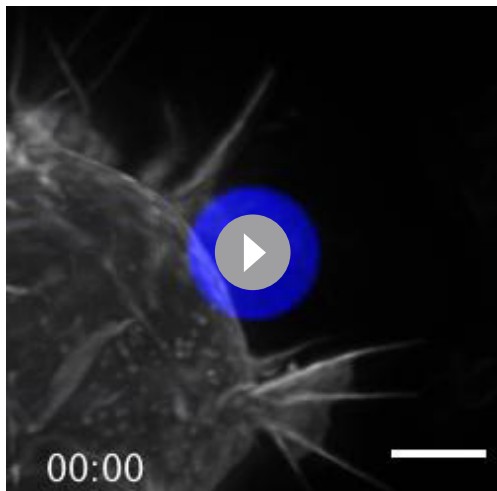

**Video 3.** RAW macrophage ingesting stiffer deformable acrylamide-*co*-acrylic acid-microparticle (DAAMP) shows similar phagocytic force-induced deformation patterns by lattice light-sheet microscopy. Maximum intensity projection of RAW macrophage transfected with mEmerald-Lifeact (gray) ingesting DAAM-particles (9 µm, 6.5 kPa) (blue) functionalized with AF647-Cadaverine, BSA, and anti-BSA IgG. Lower left time stamp: min:s. Scale bar, 5 µm.
https://elifesciences.org/articles/68627/figures#video3

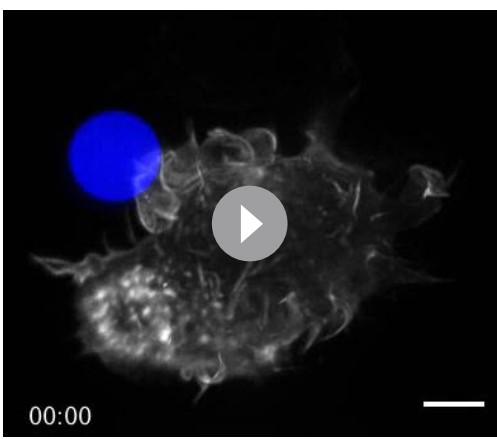

**Video 4.** Deformable acrylamide-*co*-acrylic acid-microparticle (DAAMP) phagosome briefly accumulates F-actin, imaged by lattice light-sheet microscopy. Maximum intensity projection of RAW macrophage transfected with mEmerald-Lifeact (gray) ingesting DAAM-particles (9 µm, 6.5 kPa) (blue) functionalized with AF647-Cadaverine, BSA, and anti-BSA IgG. Lower left time stamp: min:s. Scale bar, 5 µm.
https://elifesciences.org/articles/68627/figures#video4

interestingly, less F-actin accumulation in the cup rim (*Figure 1—figure supplement 2*).

To investigate how phagocytic forces change during phagocytic progression, we arranged cups in order by the fraction of their particle surface engulfed, which allowed us to reconstruct phagocytic engulfment over time from fixed cell images (*Figure 2g* and *Figure 2—figure supplement 1*). Since DAAMPs cause little optical distortion, measurable features of the phagocytic cups could be analyzed independent of cup orientation and engulfment stage (*Vorselen et al., 2020b*). We found no marked accumulation of cups at any specific stage, suggesting no bottleneck or rate-limiting steps, which had been previously reported around 50% engulfment (*van Zon et al., 2009*). However, we did observe a strong increase in global target deformation, measured as the inverse of target sphericity, with phagocytic progression (Spearman's $\rho$ = –0.62, p = 5.0 × 10$^{-8}$) (*Figure 2h*). This decrease in target sphericity was, at least partially, due to a 4% ± 1% (p = 1.9 × 10$^{-7}$) average increase in DAAMP elongation along the phagocytic axis (*Figure 2h*), which is consistent with constriction orthogonal to the phagocytic axis. Direct analysis of target constriction and F-actin peak intensity for each phagocytic cup (*Figure 2i*, *Figure 2—figure supplement 1*) revealed an apparent contractile ring in almost all (~96%) cups. The location of this actin contractile ring along the phagocytic axis correlated extremely well with phagocytic progression ($\rho$ = 0.93, p = 3.2 × 10$^{-29}$) and led to target constriction increasing from ~80 nm in early-stage (fraction engulfed <40%) to ~210 nm in late-stage cups (fraction engulfed >70%). This is a direct effect of increasing normal forces at the cup rim (*Figure 2j*). Strikingly, in early stages of phagocytosis, net pulling (or outward normal) forces were observed throughout the phagocytic cup and particularly at the base (~3 nN total force, > 100 Pa tensile stresses) (*Figure 2j*), whereas in late-stage phagocytosis strong net compressive stresses were observed.

## Arp2/3-mediated actin polymerization drives force generation throughout phagocytosis, whereas myosin-II powers cup closure

The striking observations of target constriction becoming more pronounced later in engulfment inspired us to consider distinct contributions of actin assembly and myosin-mediated contractility to force generation during phagocytosis. To separate the effects of these two actin-dependent processes, we inhibited Arp2/3-mediated and formin-mediated actin polymerization, as well as myosin-II activity using the small molecule inhibitors CK666, SMIFH2, and Blebbistatin,

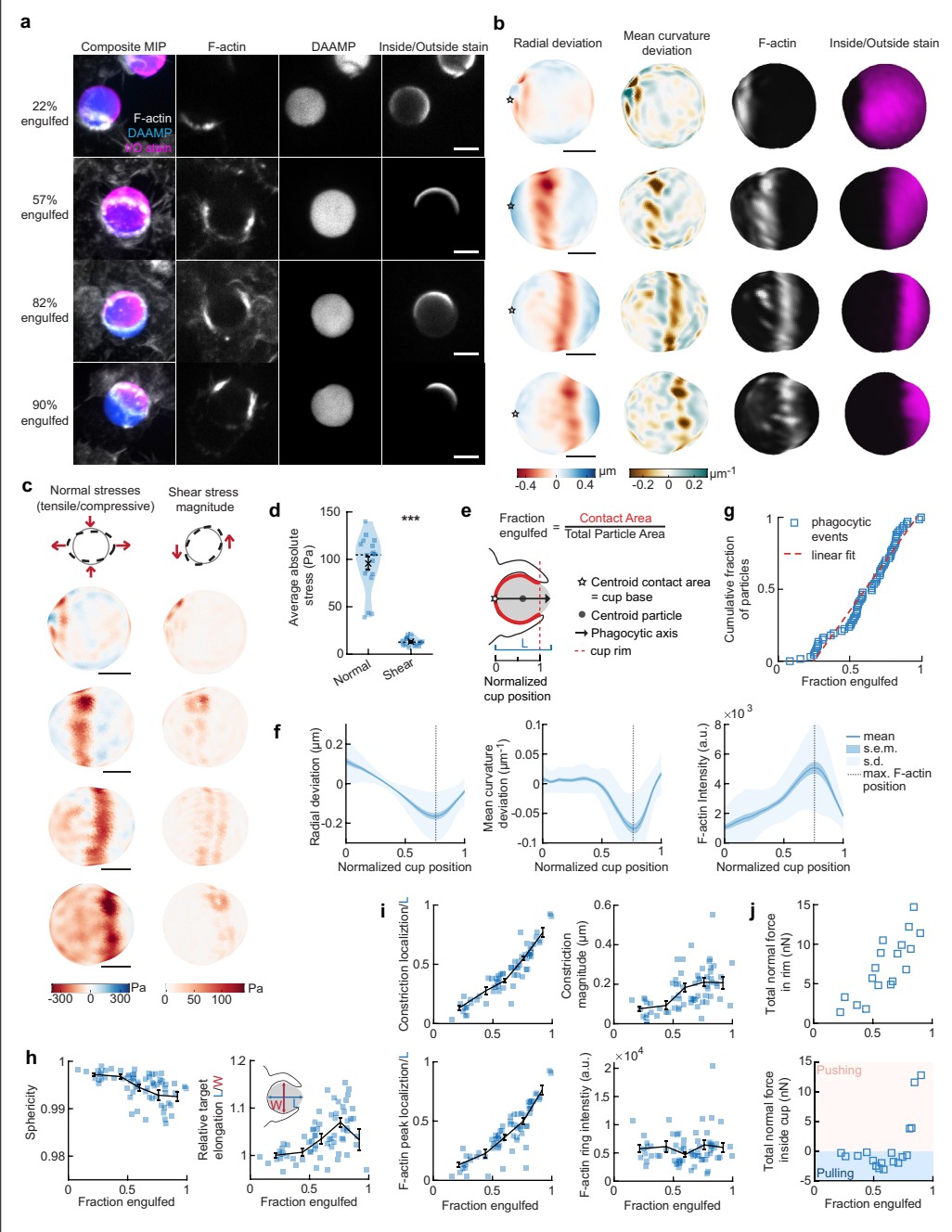

**Figure 2.** Phagocytic forces include strong actin-mediated constriction and increase with phagocytic progression.
(**a**) Confocal images of fixed RAW macrophages phagocytosing soft deformable acrylamide-*co*-acrylic acid
micro (DAAM)-particles functionalized with IgG, and AF488-Cadaverine for visualization. Cells were stained
for F-actin, and particles with a fluorescent secondary antibody to reveal the exposed surface. Left column:
composite maximum intensity projections (MIP) of confocal z-stacks, second to fourth column: single confocal
slices through particle centroid. Scale bar, 5 μm. (**b**) 3D shape reconstructions of deformable acrylamide-*co*-
acrylic acid-microparticle (DAAMP) in (**a**) revealing detailed target deformations induced during phagocytosis and
localization of F-actin over the particle surface. Stars mark the base of the phagocytic cup, and cups are aligned
with the phagocytic axis (see **e**) from left to right. Scale bars, 3 μm. (**c**) Normal and shear stresses inferred from the
shape deformations of the targets in (**a,b**). Negative normal forces denote (inward) pushing forces. (**d**) Averages
of absolute magnitudes of normal and shear stresses in phagocytic cups (n = 18). Violin plots show individual
phagocytic events (blue markers), mean (black cross) and median (dashed line). *Two-sided Wilcoxon rank sum
test: p = 2.0 × 10⁻⁴. (**e**) Schematic representation of phagocytic parametrization. Normalized cup position indicates

*Figure 2 continued on next page*

*Figure 2 continued*

the position along the phagocytic axis relative to the rim of the cup, with 0 the cup base and 1 the rim of the phagocytic cup. (**f**) Average profiles of target deformation and F-actin intensity along the phagocytic axis, where 0 and 1 are the cup base and rim, respectively. Signals were first processed on a per-particle basis by averaging over the surface along the phagocytic targets (in 30 bins). Only targets beyond 40% engulfment were included (54 out of 68 events in total). (**g**) Cumulative distribution function of the engulfment stage of randomly selected phagocytic events before completion of engulfment (n = 68). Dashed red line indicates a linear fit. (**h**) Target sphericity and elongation dependence on phagocytic stage. Blue squares indicate individual measurements, black lines indicate averages within five bins. Middle graph inset schematic shows how relative elongation was determined. (**i**) Analysis of the radially symmetric component of particle deformation and F-actin fluorescence along the phagocytic axis for all phagocytic events (n = 68). Marker and line styles as in (**h**). (**j**) Analysis of forces in the contractile ring at the cup rim and throughout the remainder of the cup for 18 cups selected for force analysis. All error bars indicate s.e.m. unless indicated otherwise. Raw data are available in *Figure 2—source data 1* and raw images are available on a FigShare repository (*Barger et al., 2021a*).

The online version of this article includes the following source data and figure supplement(s) for figure 2:

**Source data 1.** Numeric data for *Figure 2d,f–i*.

**Figure supplement 1.** Automated image analysis reveals deformation, F-actin localization, and inside/outside (I/O) stain for 68 phagocytic events.

---

respectively (*Figure 3a*, *Figure 3—figure supplements 2–5*). Although SMIFH2 was recently reported to exhibit off-target effects, most notably myosin-II inhibition (*Nishimura et al., 2021*), we observed markedly different behaviors upon SMIFH2 treatment compared to direct myosin-II inhibition using Blebbistatin (*Figure 3—figure supplement 5*). Target deformation analysis and force calculations revealed that target constriction was diminished upon inhibition of the Arp2/3 complex and myosin-II activity, while formin inhibition had a relatively modest effect (*Figure 3c,d*, *Figure 3—figure supplements 2 and 5*). The loss of target constriction coincided with a strong reduction (~40%) in F-actin accumulation at the rim of the cup, as well as a 50% broadening of the typical narrow (~2 µm) F-actin band observed in the DMSO control (*Figure 3d,e*, *Figure 3—figure supplements 2 and 5*). Of note, upon myosin-II inhibition alone, the loss of F-actin at the rim of the cup was complemented by a small, but significant (p = 0.04), increase in F-actin density at the base of the cup (*Figure 3d,f*, *Figure 3—figure supplement 5*). This observation suggests that myosin-II may be promoting actin disassembly at the base of the phagocytic cup during internalization, similar to the role of myosin-II in disassembling the F-actin network at the cell rear during cell motility (*Wilson et al., 2010*). To understand the implications of these distinct force generation profiles on phagocytic efficiency, we challenged RAW macrophages to engulf 9 µm 1.4 kPa DAAMPs under drug treatment. Similar to previous reports using large polystyrene particles (*Rotty et al., 2017*), inhibition of the Arp2/3 complex strongly reduced phagocytosis after 15 min and uptake remained low after longer incubation (*Figure 3g*, *Figure 3—figure supplement 2*). Meanwhile, uptake was not significantly affected by Blebbistatin treatment and slightly reduced upon SMIFH2 treatment (*Figure 3g*, *Figure 3—figure supplement 5*).

We then investigated whether the activity of these molecular players may be associated with specific phagocytic stages. Our analysis revealed a significant change in the observed distribution of cup stages upon myosin-II inhibition, but not Arp2/3 or formin perturbation, compared to DMSO-treated control cells (*Figure 3h* and *Figure 3—figure supplement 5*). Specifically, in the Blebbistatin-treated cells, we found an >6 -fold enrichment of cups that were beyond 90% engulfment, but not yet closed (p = $1.9 \times 10^{-4}$). This high prevalence of late-stage phagocytic cups suggests a specific role for myosin-II in cup closure. Throughout phagocytosis, general particle deformations, as measured by the decrease in target sphericity, were strongly reduced upon both CK666 and Blebbistatin treatment (*Figure 3i*). Inhibition of formins generally reduced target deformations, but also increased the cell-to-cell variability in particle deformation, suggesting that formins may play a role in fine-tuning phagocytic force production (*Figure 3—figure supplement 5*). Whereas overall target deformations were reduced in all stages of phagocytosis upon Arp2/3 inhibition, myosin-II inhibition only significantly affected phagocytic force generation at later stages, after 50% engulfment (*Figure 3i*). A similar effect was observed when quantifying target constriction and target elongation specifically (*Figure 3j*, *Figure 3—figure supplement 2*). This analysis strongly suggests that there is a specific role for myosin-II in late-stage phagocytosis despite previous data that shows myosin-II is dispensable

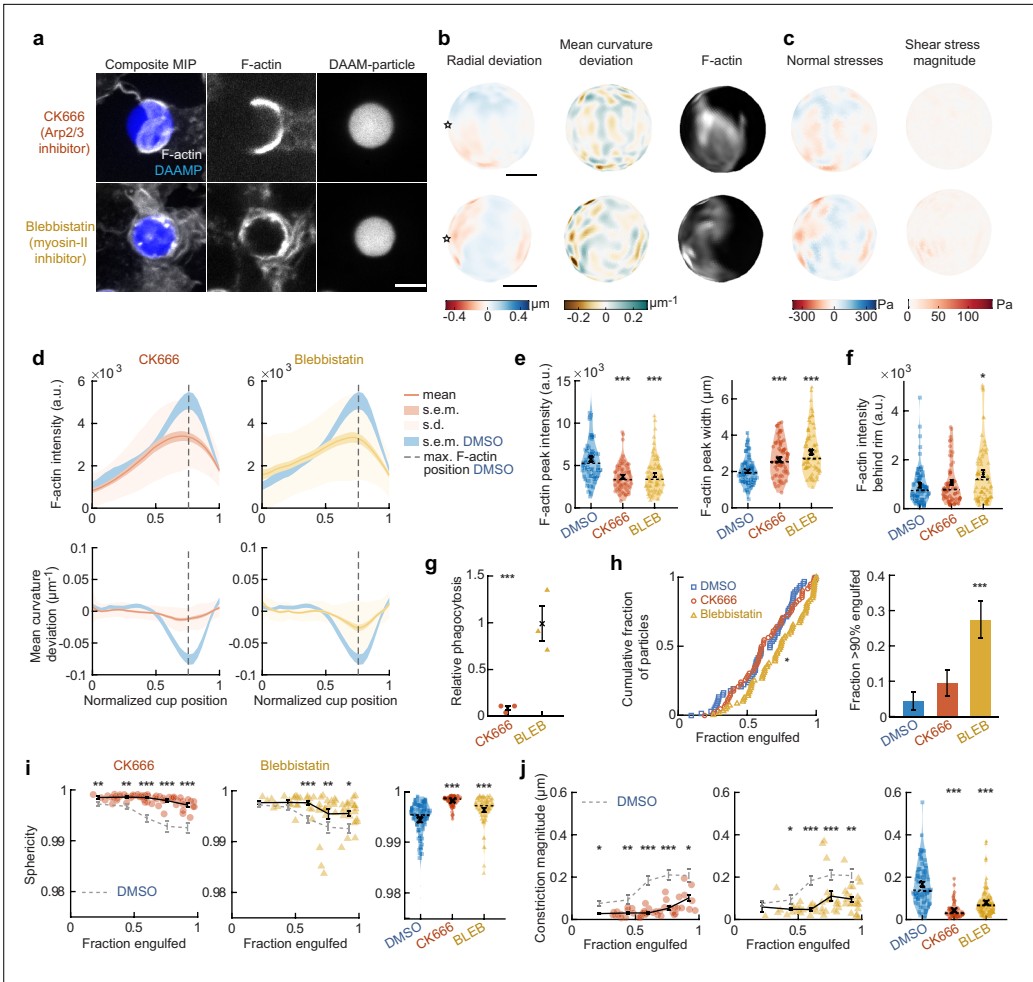

**Figure 3.** Arp2/3-mediated actin polymerization and myosin-II have distinct roles in phagocytic force generation and progression. (**a**) Confocal images of drug-treated fixed RAW cells phagocytosing deformable acrylamide-*co*-acrylic acid micro (DAAM)-particles functionalized with IgG and AF488-Cadaverine for visualization. Cells were treated with DMSO, CK666 (150 μM), and Blebbistatin (15 μM) for 30 min prior to phagocytic challenge. Each target is approximately 60% engulfed. Fixed cells were stained for F-actin, and particles were labeled with a fluorescent secondary antibody to reveal the exposed surface. Left column: composite maximum intensity projections (MIP) of confocal z-stacks, second to third column: single confocal slices through particle centroid. Scale bar, 5 μm. (**b**) Particle shape reconstructions from (**a**) revealing cell-induced target deformations and localization of F-actin over the particle surface. Stars mark the base of the phagocytic cup, cups are aligned with the phagocytic axis (see *Figure 2e*) from left to right. Scale bars, 3 μm. (**c**) Normal and shear stresses derived from target deformations. Negative normal forces denote (inward) pushing forces. (**d**) Average profiles of target deformation and F-actin intensity along the phagocytic axis. Signals were first processed on a per-particle basis by averaging over the surface along the phagocytic targets in 30 bins. Targets before 40% engulfment were excluded. (**e, f**) Violin plots showing individual phagocytic events (colored markers), mean (black cross), and median (dashed line). (**e**) F-actin peak intensity and band width. (**f**) F-actin intensity in the cup (behind the rim), measured right (3 μm) behind the main peak for each particle. (**g**) Phagocytic efficiency upon drug treatment evaluated as the number of internalized particles divided by the total number of cell-associated particles. Uptake was evaluated 15 min after addition of particles and normalized to internalization by DMSO-treated cells. Three independent experiments were performed where 80–200 particles were measured per condition for each experiment. ***p = 0.0007 (t-test result for hypothesis, mean = 1). (**h**) Upper panel, cumulative distribution function of the engulfment stage of randomly selected phagocytic events before completion of engulfment (*n* = 68, 63, 73 respectively) from three independent experiments. Two sample Kolmogorov-Smirnov test was used (p = 0.016*). Lower panel, fraction late-stage cups. Error bars indicate st.d. estimated by treating phagocytosis as a Bernoulli process. Fisher's exact test was used to compare fractions (p = 1.9 × 10⁻⁴)***. (**i**) Sphericity and (**j**) constriction magnitude of DAAM-particle changes with phagocytic progression upon drug treatment. Colored markers indicate individual events,

*Figure 3 continued on next page*

*Figure 3 continued*

black lines indicate averages of five bins. Right column, violin plots of all events. Marker and line styles as in (**e**). All statistical tests were two-sided Wilcoxon rank sum test comparing with the DMSO control (gray) over the same bin with significance levels: p < 0.05\*; p < 0.01\*\*; p < 0.001\*\*\*, unless otherwise indicated. All error bars indicate s.e.m. unless indicated otherwise. Raw data are available in *Figure 3—source data 1* and raw images are available on a FigShare repository (*Barger et al., 2021a*).

The online version of this article includes the following source data and figure supplement(s) for figure 3:

**Source data 1.** Numeric data for *Figure 3d–j* and *Figure 3—figure supplement 1a–e*.

**Figure supplement 1.** Actin organization and target deformation along the phagocytic axis are affected by perturbation of actomyosin activity.

**Figure supplement 2.** Automated image analysis reveals deformation, F-actin localization, and inside/outside (I/O) stain for 63 phagocytic events in CK666-treated cells.

**Figure supplement 3.** Automated image analysis reveals deformation, F-actin localization, and inside/outside (I/O) stain for 75 phagocytic events in Blebbistatin-treated cells.

**Figure supplement 4.** Automated image analysis reveals deformation, F-actin localization, and inside/outside (I/O) stain for 55 phagocytic events in SMIFH2-treated cells.

**Figure supplement 5.** SMIFH2 treatment has modest effect on force generation and F-actin distribution during phagocytosis.

---

for phagocytosis (*Figure 3g*; *Jaumouillé et al., 2019*; *Olazabal et al., 2002*; *Rotty et al., 2017*). The strong effect of Arp2/3 inhibition on phagocytic efficiency and target deformations throughout phagocytosis, and the lack of effect on the distribution of cups-in-progress, suggests that branched actin assembly plays an important role throughout phagocytic progression.

## Actin-based protrusive teeth drive target constriction and are mechanosensitive

Based on our observations of discrete spots of inward deformation using MP-TFM (*Figures 1a, b, 2a and b*), and the significant reduction in target deformation after treatment of cells with the Arp2/3 inhibitor CK666, we hypothesized that these local deformations were the result of actin-based protrusions pushing against the surface of the phagocytic target. Indeed, we frequently observed actin-rich puncta that appeared as oblong or triangular tooth-like projections locally indenting the target surface along the internal rim of the phagocytic cup (*Figure 4a*) and sometimes deeper within the cup (*Figure 4b*). Similar actin 'teeth' were formed by primary murine BMDMs, bone marrow-derived dendritic cells (BMDCs), and HL-60 human neutrophils when challenged with IgG-functionalized DAAMPs, suggesting that these structures are a common feature of phagocytosis (*Figure 4c*).

To investigate the nature and biological function of these actin 'teeth' more carefully, we identified them on individual particles based on their protrusive nature and high F-actin intensity (*Figure 4—figure supplement 1*). According to these criteria, teeth were found in almost all phagocytic cups, with ~10 distinct teeth per cup (*Figure 4d*). Typically ~1 µm in diameter, and protruding ~200 nm into the 1.4 kPa DAAMP targets, they resembled podosomes in size and protrusive nature (*Labernadie et al., 2014*). Cells treated with the Arp2/3 inhibitor CK666, and to a lesser extent cells treated with the formin inhibitor SMIFH2, exhibited a reduction in the number of actin teeth per cup (80% and 40% reduction, respectively) compared to control cells treated with DMSO (*Figure 4d*, *Figure 3—figure supplement 5*). This result suggests that, like podosomes (*van den Dries et al., 2019b*), target-deforming phagocytic teeth include both Arp2/3- and formin-nucleated actin filaments. Surprisingly, a strong decrease (~50%) in the number of actin teeth was also observed upon myosin-II inhibition. For all treatments, the reduced number of individual teeth that still formed were remarkably similar to those formed by control cells. Although tooth size and depth were reduced upon CK666 or Blebbistatin treatment, the effect size was small (<15%), suggesting that 'teeth' are resilient structures that, once formed, have well-defined properties (*Figure 4d*).

LLSM allowed us to track the dynamics of actin teeth during phagocytosis (*Figure 4e*), which revealed clear forward movement over the target surface. A few teeth that we initially detected near the rim stayed in place, remaining where they had assembled during early-stage phagocytosis, suggesting that teeth located deeper within the phagocytic cup (as observed in the fixed cell images)

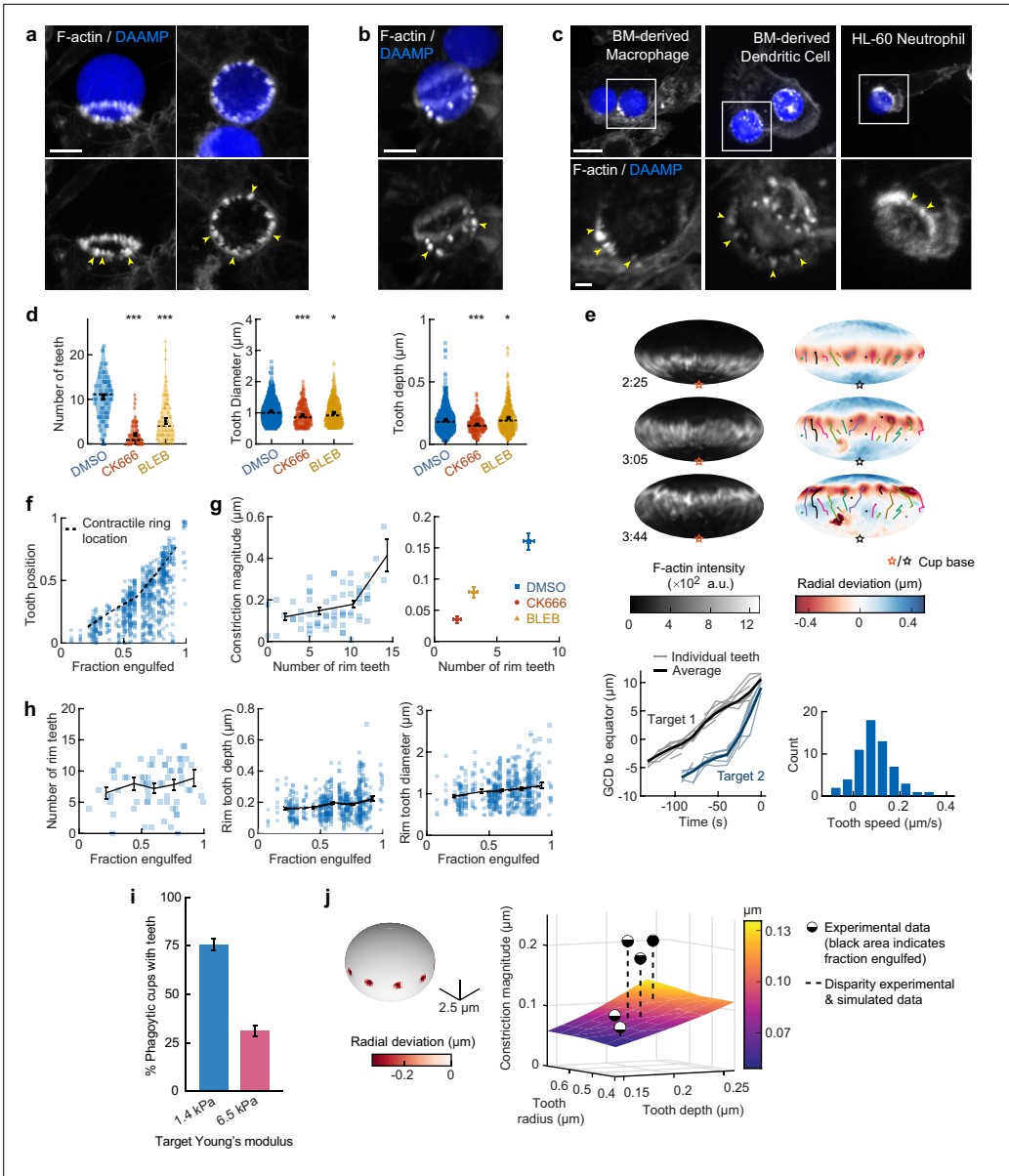

**Figure 4.** Actin-based teeth are dynamic, and likely interconnected, structures whose protrusive activity contributes to target constriction. (**a**,**b**) RAW macrophages were fed 1.4 kPa deformable acrylamide-*co*-acrylic acid-microparticles (DAAMPs) and stained for F-actin. Images represent maximum intensity projections of confocal z-stacks. Yellow arrows point to actin-based 'teeth'. (**a**) Representative images of teeth at the rim of the phagocytic cup deforming the target. Scale bar, 5 μm. (**b**) Example of actin-based teeth observed within the phagocytic cup. Scale bar, 5 μm. (**c**) Primary murine bone marrow-derived macrophage (BMDM), as well as primary bone marrow-derived dendritic cell (BMDC) and HL-60 neutrophil-like cells also form actin teeth in response to DAAMP internalization. Scale bar, 5 μm. Zoom scale bar, 2 μm. (**d**) CK666 and Blebbistatin treatments reduce formation of actin-based teeth within the phagocytic cup. Teeth size and shape are also modestly affected. Violin plots with individual particles (colored markers), means (black cross), median (dashed line). Stars indicate Wilcoxon rank sum test results comparing with the DMSO control with significance levels: p < 0.05*; p < 0.001***. (**e**) Manually tracked actin teeth trajectories from DAAMP internalization imaged by lattice light-sheet microscopy (LLSM). Particle surface is shown using Mollweide projection (*Figure 2—figure supplement 1*). Three time points of a single phagocytic event are shown, with different colors representing unique teeth. Circles indicate current or final position of a tracked tooth. Lines connect the previous positions of tracked teeth. Lower left: great circle distance (GCD) of teeth to the equator for two events. Target 1 is visualized above, time 0 corresponds to the time at which engulfment was completed. Lower right: distribution of tooth speeds with average 0.094 ± 0.08 μm/s (=5.6 μm/min) from three phagocytic events (60 teeth in total from targets 1 and 2 in this panel, and from *Figure 4—figure*

*Figure 4 continued on next page*

*Figure 4 continued*

*supplement 1*). Tooth speeds were averaged over the trajectory of individual teeth. (**f**) Teeth are mostly located at the rim of the cup. Markers represent individual teeth ($n$ = 716). (**g**) Constriction magnitude correlates with the number of teeth with Spearman's rank correlation coefficient ($r$) = 0.42 (p = $4.4*10^{-4}$) for individual DMSO phagocytic events (left) and between drug treatments (right). (**h**) Teeth number and features change with phagocytic progression, with, from left to right, $r$ = 0.2 (p = 0.11 n.s.), $r$ = 0.17 (p = $1.0 \times 10^{-4}$), $r$ = 0.16 (p = $2.1 \times 10^{-4}$). (**i**) Phagocytic cups with actin teeth appear more frequently when cells are challenged with softer targets. RAW macrophages were challenged with 9 µm DAAMPs of 1.4 or 6.5 kPa, fixed and stained for F-actin. (**j**) Elasticity theory simulations of the relation between tooth size and depth and overall constriction magnitude. Inset shows teeth, simulated as spherical indenters on a spherical target. Bar graph represents pooled data (n = >150 cups) from three independent experiments. Error bars indicate st.d. estimated by treating phagocytosis as a Bernoulli process. Pooled data was compared using Fisher's exact test to compare fractions (p = $1.5 \times 10^{-24}$***). All error bars indicate s.e.m. unless otherwise indicated. Raw data are available in *Figure 4—source data 1*.

The online version of this article includes the following source data and figure supplement(s) for figure 4:

**Source data 1.** Numeric data for *Figure 4d–j*, *Figure 4—figure supplement 1d–g* and *Figure 4—figure supplement 2c*.

**Figure supplement 1.** Automated teeth identification reveals how teeth number, size, and indentation correlate with overall target constriction.

**Figure supplement 2.** Indentation simulations of phagocytic teeth allow comparison of teeth number, size, and indentation with and total target constriction.

---

may have originated earlier from the cup rim and been left behind as the cup progressed (*Figure 4b*, *Figure 4—figure supplement 1*). More commonly, however, teeth moved forward with a speed of ~5.6 µm/min, similar to the previously reported values for podosome-like structures on very stiff substrates during frustrated phagocytosis (*Ostrowski et al., 2019*). Strikingly, teeth within the same phagocytic cup appeared to move in a coordinated fashion, with similar speed and direction, and even with observed collective speed changes (*Figure 4e*, *Figure 4—figure supplement 1*). This suggests that phagocytic teeth, like podosomes (*Meddens et al., 2016*; *Proag et al., 2015*), are mechanically interlinked at the mesoscale.

To test whether the actin teeth were mechanosensitive, we challenged RAW macrophages to ingest 9 µm DAAMPs of 1.4 or 6.5 kPa and fixed and stained cells to examine actin teeth formation. Of note, drug treatments had similar effects on uptake of the stiffer particles as the softer particles (*Figure 1—figure supplement 2*). Because deformations induced on the stiffer targets were smaller, our systematic identification of teeth based on protrusive activity and actin intensity could not be utilized to faithfully compare tooth number or depth between targets of varying stiffness (*Figure 1—figure supplement 2*). However, by assessing actin distribution, RAW macrophages assembled actin teeth more frequently when fed softer targets (*Figure 4i*). This is consistent with our findings that there is less overall F-actin accumulation in the cup rim on stiffer targets (*Figure 1—figure supplement 2*) and suggests that phagocytic teeth may play a role in the overall mechanosensitivity of phagocytosis (*Beningo and Wang, 2002*; *Jaumouillé et al., 2019*; *Sosale et al., 2015*; *Vorselen et al., 2020b*; *Vorselen et al., 2020a*).

Given the ring-like organization of phagocytic teeth in the cup rim, combined with their individual protrusive activity, we questioned whether they were sufficient to explain our observations of target constriction orthogonal to the phagocytic axis, or if a separate contractile mechanism is required. We first distinguished the teeth positioned at the rim of the cup (~70% of teeth), which likely contribute to target constriction, from those deeper in the cup, based on their distance from the cup rim (*Figure 4f*, *Figure 4—figure supplement 1*). We then determined whether the properties of teeth near the rim correlated with the overall target constriction. Indeed, the number of teeth per cup and tooth size correlated with overall constriction in DMSO-treated cells and between groups treated with actomyosin activity inhibitors CK666, SMIFH2, and Blebbistatin (*Figure 4g*, *Figure 4—figure supplement 1*). We further examined whether changes in the teeth could be related to increasing target constriction with phagocytic cup progression. Teeth numbers increased only slightly with phagocytic progression, which suggests that they are formed quickly early on in phagocytosis and are then typically maintained at constant numbers throughout engulfment (*Figure 4h*). Teeth size and depth increased significantly but modestly during phagocytic progression (*Figure 4h*, *Figure 4—figure supplement 1*). Elasticity

theory simulations of teeth-like indentations of a spherical target allowed us to test whether teeth protrusive activity is sufficient to explain the extent of overall target constriction in different stages of phagocytosis (*Figure 4—figure supplement 2*). Remarkably, this revealed that teeth activity is indeed sufficient to account for total target constriction in early-stage phagocytosis (<50% engulfment), but insufficient to explain the greater degree of target constriction later in the process (*Figure 4j*). This is consistent with additional myosin-II-based contractile forces in late-stage phagocytosis, as suggested by our observations using Blebbistatin.

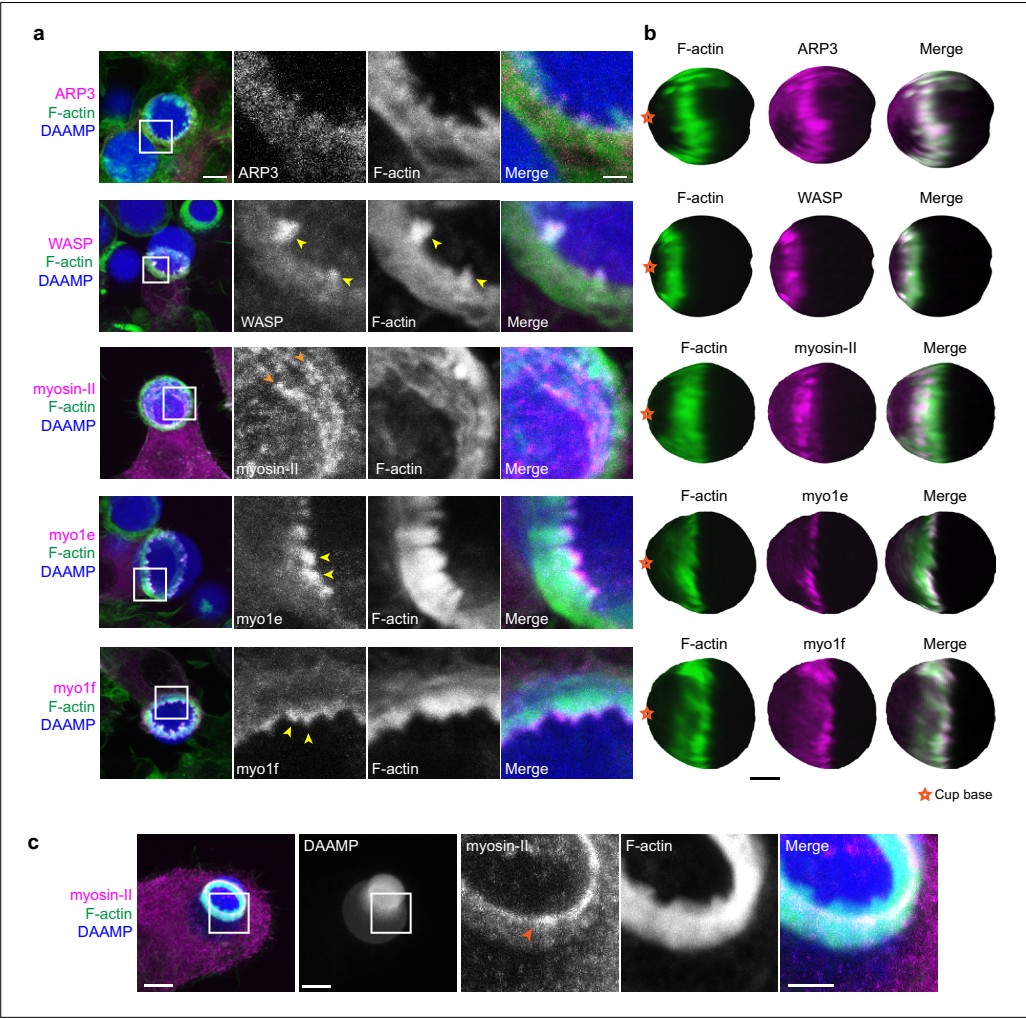

**Figure 5.** Multiple actin regulatory proteins localize to phagocytic teeth. (**a**) RAW macrophages were transfected with fluorescently tagged actin-binding proteins and challenged to ingest deformable acrylamide-*co*-acrylic acid-microparticles (DAAMPs) (11 μm, 1.4 kPa) functionalized with IgG and AF647-Cadaverine to assess localization to actin teeth (yellow arrowheads). Images are maximum intensity projections of confocal z-stacks. White boxes in leftmost panels indicate the site of the zoomed images to the right. Scale bar, 5 μm. Zoom scale bar, 1 μm. (**b**) DAAM-particle reconstructions for examples shown in (**a**) showing target deformations and localization of fluorescent proteins with respect to actin teeth. Scale bar, 3 μm. (**c**) Myosin-II condensing into thick concentric rings (marked by orange arrowheads) during late-stage phagocytosis of a highly deformed target. Images are maximum intensity projections of confocal z-stacks. Scale bar, 5 μm. Zoom scale bar, 1 μm. Raw images are available on a FigShare repository (*Barger et al., 2021b*).

The online version of this article includes the following figure supplement(s) for figure 5:

**Figure supplement 1.** Multiple actin-binding proteins localize to phagocytic teeth.

## Regulators of branched actin assembly (Arp2/3 and WASP) and myosin-I isoforms localize to actin teeth while myosin-II forms contractile rings within the phagocytic cup

Due to the resemblance of the phagocytic actin teeth to podosomes in size, protrusive activity (*Figure 4d*), and dynamics (*Figure 4e*), we naturally questioned whether these structures were similar in protein composition as well. Given technical challenges with immunohistochemical staining and MP-TFM (see Materials and methods), we transfected the RAW macrophages with fluorescently tagged proteins to assess localization relative to the actin teeth using 3D reconstructions of the DAAMPs. Consistent with our earlier results showing a decrease in the number of teeth after treating cells with the Arp2/3 inhibitor, we found that the Arp2/3 complex and its activator, WASP, colocalized with the actin teeth (*Figure 5a and b*). Moreover, cortactin and cofilin, actin-binding proteins that are frequently found in association with densely branched actin networks, also localized to the phagocytic teeth (*Figure 5—figure supplement 1*). In contrast, myosin-II often appeared distinctly behind the actin teeth in an anti-correlated fashion (*Figure 5b*). In particular, rings composed of myosin-II filaments could be seen within the phagocytic cup (*Figure 5a*). In cups that were almost closed, myosin-II coalesced into a ring behind the actin rim (*Figure 5—figure supplement 1*) and clearly localized to sites of target constriction in cases of extreme target deformation (*Figure 5c*). Meanwhile, the two long-tailed myosin-I isoforms, myo1e and myo1f, localized specifically to the tips of the actin teeth, consistent with our previous observations (*Figure 5a*; *Barger et al., 2019*).

Given both recent and older observations identifying adhesion adaptor proteins at the phagocytic cup (*Beningo and Wang, 2002*; *Greenberg et al., 1990*), we were particularly interested in the localization of paxillin and vinculin. In comparison to the branched actin-binding proteins, both paxillin and vinculin localized behind the phagocytic teeth in a punctate-like pattern (*Figure 5—figure supplement 1*). Altogether, these studies support a model whereby actin teeth are composed of branched actin filaments guided by myosin-I motor proteins. The localization of myosin-II and paxillin/vinculin behind the teeth suggests that these proteins may play a role in the potential interconnection of the teeth, similar to podosomes on 2D surfaces (*Meddens et al., 2016*; *van den Dries et al., 2019b*).

## Contractile activity may enable resolution of phagocytic conflicts via partial target eating ('nibbling') or forfeit of uptake ('popping')

While we have found that target constriction is a signature mechanical feature of FcR-mediated phagocytic progression, it is unclear what functional role target constriction might play during phagocytosis, since actin-driven membrane advancement along the target surface should in principle be sufficient for internalization (*Herant et al., 2006*; *Tollis et al., 2010*). In addition to the many successful internalization events we observed using LLSM, we also observed some strikingly different target encounters in which RAW macrophages assembled large amounts of F-actin, only to squeeze futilely at the base of the target without completing engulfment. In these cases, the contractile activity resulted in dramatic deformations and even dumbbell-like appearance of the target (*Figure 6a and b*, *Video 5*). In addition to this kind of internalization failure by single cells, we also observed incidents where two macrophages engaged one DAAMP target. Similar to previous observations of red blood cells being squeezed into multilobed shapes when attacked by two macrophages simultaneously (*Swanson et al., 1999*), these conflicts were also observed using primary BMDMs challenged with DAAMPs (*Video 6*). Although the polymeric targets used in this study prohibit partial target eating because each particle is effectively one single crosslinked macromolecule that cannot easily be severed by cell-exerted forces, this behavior may be reminiscent of trogocytosis, the process which has been observed during immune cell attack of cancer cells whereby phagocytes ingest small bits of their target (*Matlung et al., 2018*; *Morrissey et al., 2018*; *Velmurugan et al., 2016*). By imaging RAW macrophages transfected with GFP-tagged non-muscle myosin-IIa, we observed highly enriched rings of myosin-II signal at DAAMP deformations during attempts of partial target eating (*Figure 6c*, *Video 7*). Target encounters involving extreme deformations of the DAAMP also revealed the existence of a 'popping' mechanism that could lead to a sudden release of the target (*Video 8*) or, conversely, a sudden completion of engulfment (*Video 9*, *Figure 6—figure supplement 1*). During such events, targets were first gradually deformed to a dumbbell-like shape, followed by a sudden translocation of the particle, as well as an immediate recovery of its original spherical shape and volume (*Figure 6d*, *Video 7*). The rapid timescale of this process suggests that it is likely purely mechanical, representing an elastic recoil

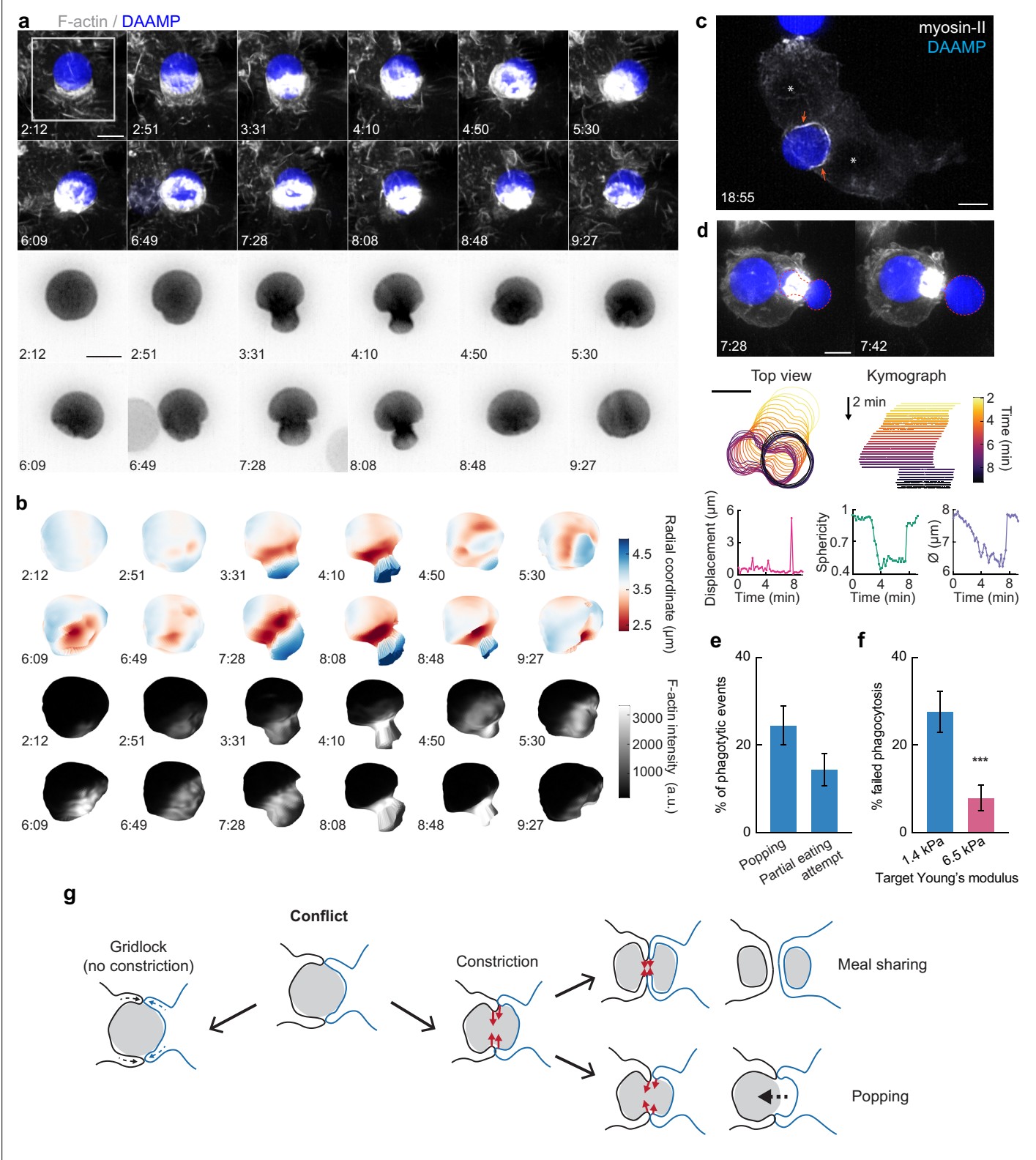

**Figure 6.** Contractile activity may enable resolving conflicts by partial target eating and a popping release mechanism. (**a**) Maximum intensity projections (MIP) of lattice light-sheet microscopy (LLSM) stacks showing failed internalization attempt of RAW macrophage with IgG-functionalized 1.4 kPa deformable acrylamide-*co*-acrylic acid-microparticle (DAAMP). Zoomed images of the area marked by the white box showing only the DAAMP channel (inverse grayscale LUT) are shown below. (**b**) 3D reconstructions showing both the particle shape and F-actin signal over the particle surface

*Figure 6 continued on next page*

*Figure 6 continued*

corresponding to the event in (**a**). Scale bar, 3 µm. (**c**) Maximum intensity projections (MIP) of LLSM stacks showing myosin-II accumulation (orange arrows) during a partial eating event. RAW macrophages (marked by *) were transfected with EGFP-NMMIIA (non-muscle myosin-IIa) to label myosin-II and challenged with IgG-functionalized 1.4 kPa DAAMP. (**d**) Top: MIPs of LLSM stacks of RAW macrophage suddenly releasing heavily deformed target. Red dashed line outlines DAAMP. Middle: particle position and outline (left), and kymograph of particle position (right). Bottom: particle displacement, sphericity, and apparent diameter over time of the same event shows the sudden nature of the release. (**e**) Sudden forfeit by a popping mechanism and attempted partial eating are common for 1.4 kPa targets, with ~24% and ~ 14% occurrence of all phagocytic events, respectively. (**f**) Percentage of failed phagocytic events is dependent on particle rigidity. Data from two to three independent experiments was pooled ($n$ = 89, 91 phagocytic events) and compared using Fisher's exact test (p = $7.4 \times 10^{-4}$). (**g**) Schematic representation of the multiple ways in which target constriction may enable resolving macrophage conflicts in which two cells attempt a single target. All scale bars are 5 µm, unless otherwise indicated. All error bars indicate s.d. estimated by treating phagocytosis as a Bernoulli process. Raw data are available in *Figure 6—source data 1*.

The online version of this article includes the following source data and figure supplement(s) for figure 6:

**Source data 1.** Numeric data for *Figure 6d–f* and *Figure 6—figure supplement 1b*.

**Figure supplement 1.** Target squeezing can result in the target popping into the phagocytic cup.

of the DAAMP. Importantly, these encounters were rather common, with the attempted partial eating attempts (~14%) and popping (~24%) making up almost 40% off all recorded events (*Figure 6e*). Furthermore, such events, and specifically popping, were mechanosensitive and occurred much less frequently for stiffer 6.5 kPa targets (~1%, $n$ = 89, p = $1.5 \times 10^{-6}$), resulting in the overall more frequent failure of phagocytosis for soft particles (*Figure 6f*).

## Discussion

By combining LLSM and MP-TFM, we report here a detailed analysis of the mechanical progression of phagocytosis and the contributions of several key molecular players. We have discovered that FcR-mediated phagocytosis occurs through a unique mechanism in which normal forces dominate over shear forces in the cell-target interaction. This is in contrast to the current view that the phagocytic cup is equivalent to the leading edge of a migrating cell, where shear forces typically predominate at the cell-substrate interface (*Case and Waterman, 2015*; *Legant et al., 2013*). In addition, the fast forward movement of actin teeth, which underlie target constriction, concomitant with phagocytic cup progression across the target is in stark contrast to lamellipodial focal adhesion complexes, which are fixed relative to the substratum (*Case and Waterman, 2015*). The strong target constriction recently observed in complement-mediated phagocytosis by peritoneal macrophages in vitro (*Walbaum et al., 2021*)

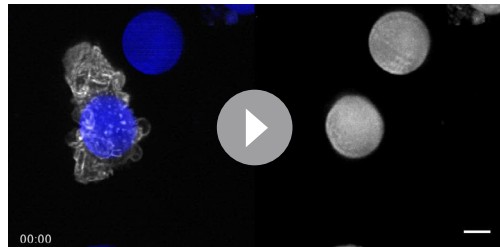

**Video 6.** Primary bone marrow-derived macrophage (BMDM) partakes in deformable acrylamide-*co*-acrylic acid-microparticle (DAAMP) meal sharing imaged by lattice light-sheet microscopy. Murine BMDMs transfected with mEmerald-Lifeact (gray) were fed DAAM-particles (11 µm,1.4 kPa) (blue) functionalized with TRITC-Cadaverine, BSA, and anti-BSA IgG. Merged maximum intensity projections (left) with single DAAMP channel in gray (right) to highlight target deformation. Transfected macrophage attempts to bite DAAMP in half with second untransfected macrophage on the other end, whose presence is implicated by local deformations on the side of the particle not in contact with the transfected cell. Time stamp: min:s. Scale bar, 5 µm.

https://elifesciences.org/articles/68627/figures#video6

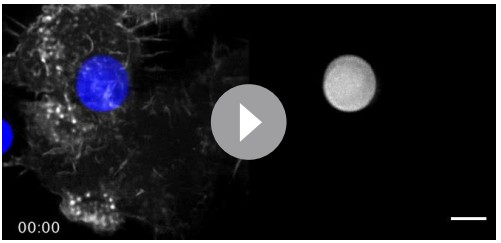

**Video 5.** Failed deformable acrylamide-*co*-acrylic acid-microparticle (DAAMP) phagocytosis imaged by lattice light-sheet microscopy. Maximum intensity projections of RAW macrophages transfected with mEmerald-Lifeact (gray) challenged with DAAM-particles (9 µm,1.4 kPa) (blue). Merged images (left) with single DAAMP channel in gray (right) to highlight target deformations. Time stamp: min:s. Scale bar, 5 µm.

https://elifesciences.org/articles/68627/figures#video5

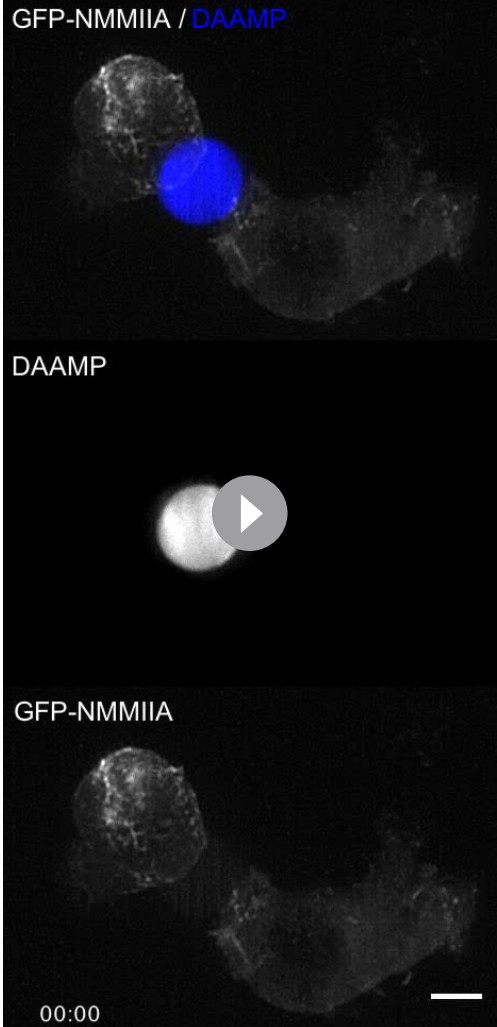

GFP-NMMIIA / DAAMP

DAAMP

GFP-NMMIIA

00:00

**Video 7.** Contractile activity involved in phagocytic conflict marked by myosin-II. Maximum intensity projections of RAW macrophages transfected with EGFP-NMMIIA (non-muscle myosin-IIa) (gray) challenged with deformable acrylamide-*co*-acrylic acid-micro (DAAM)-particles (9 μm,1.4 kPa) (blue). Time stamp: min:s. Scale bar, 5 μm.
https://elifesciences.org/articles/68627/figures#video7

their presence in multiple phagocytic cell types (*Figure 4c*) and during complement-mediated phagocytosis (*Ostrowski et al., 2019*) suggests a common role in internalization. Structures with podosome-like features, albeit resistant to CK666 treatment, have also been reported on closed phagosomes containing IgG or serum-opsonized polystyrene targets (*Tertrais et al., 2021*). We find that phagocytic actin teeth are podosome-like in protein composition, consisting of mostly branched F-actin and actin regulatory proteins. A previous study showed that the Arp2/3 complex was not required for FcR-mediated ingestion of

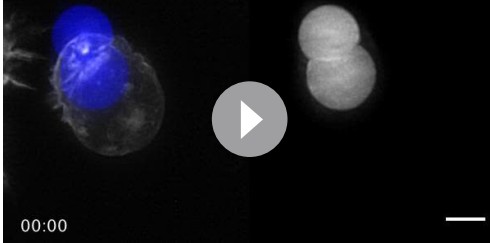

00:00

**Video 8.** Contractile activity leads to sudden forfeit of phagocytic target. Lattice light-sheet microscopy (LLSM) maximum intensity projections of RAW macrophage expressing mEmerald-Lifeact (gray) challenged with deformable acrylamide-*co*-acrylic acid-micro (DAAM)-particles (DAAMPs) (9 μm,1.4 kPa) (blue). Transfected cell attempts to internalize second DAAMP target leading to meal sharing event with second, untransfected cell. Dramatic biting of the DAAMP in two leads to the sudden forfeit of the phagocytic target. Merged images (left) with single DAAMP channel in gray (right) to highlight target deformations. Time stamp: min:s. Scale bar, 5 μm.
https://elifesciences.org/articles/68627/figures#video8

and in phosphatidylserine-mediated phagocytosis by epithelial cells in zebrafish embryos (*Hoijman et al., 2021*) suggests that strong normal forces and target constriction are likely a general feature of phagocytosis.

We show that these normal forces are primarily generated by protrusive phagocytic actin 'teeth' and myosin-II contractility, which make distinct contributions to target constriction. While actin-based puncta have been previously observed during phagocytosis of stiffer polystyrene beads (*Barger et al., 2019*; *Ostrowski et al., 2019*),

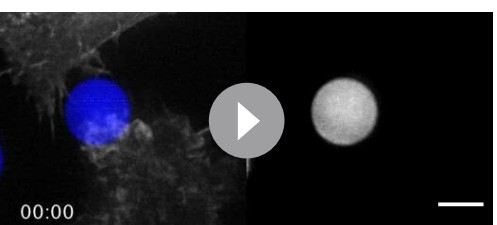

00:00

**Video 9.** Contractile activity leads to sudden completion of phagocytic internalization. Lattice light-sheet microscopy (LLSM) maximum intensity projections of RAW macrophage expressing mEmerald-Lifeact (gray) ingesting deformable acrylamide-*co*-acrylic acid-micro (DAAM)-particles (DAAMPs) (9 μm,1.4 kPa) (blue). Contractile activity on the DAAMP leads to sudden 'popping' of target toward the cell to complete ingestion. Concentrated F-actin ring appears to lag behind this event. Merged images (left) with single DAAMP channel in gray (right) to highlight target displacement. Time stamp: min:s. Scale bar, 5 μm.
https://elifesciences.org/articles/68627/figures#video9

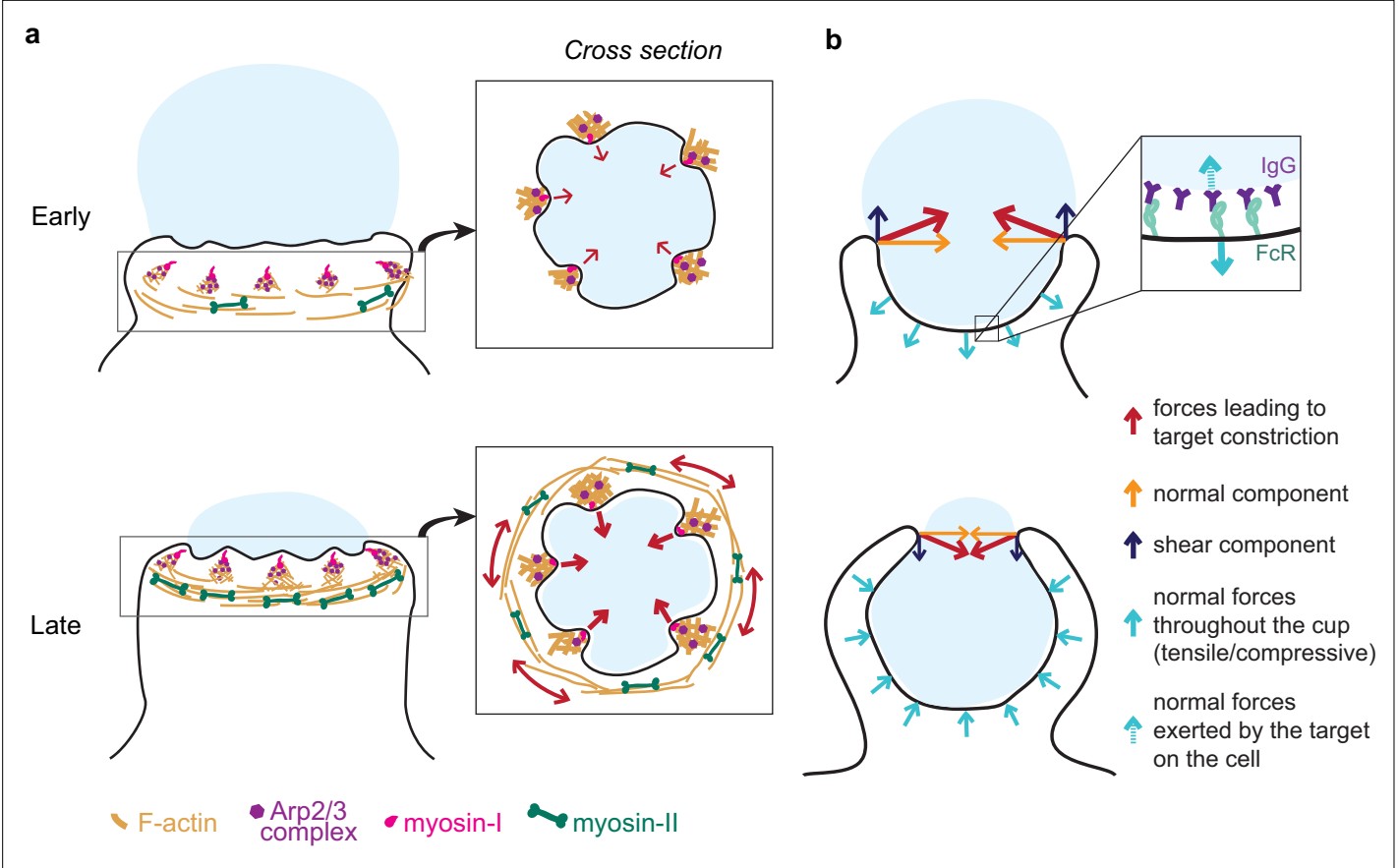

**Figure 7.** Model of the molecular players and force balance during phagocytosis. (**a**) Graphical model: Arp2/3-mediated actin teeth, guided by myosin-I motors, and organized in a ring drive phagocytic progression and inward target deformation. Teeth are potentially interconnected through myosin-II filaments located behind the actin teeth. (**b**) Ring-like target constriction by protrusive actin teeth and myosin-II activity (red arrows) can be decomposed in forces orthogonal to the phagocytic axis, which are balanced within the ring (yellow arrows) and along the phagocytic axis, which result in a net force exerted by the ring (purple arrows). The local target geometry at the protruding rim of the cup determines the direction of the net force. Before 50% engulfment, this force points outward and is balanced by pulling forces throughout the base of the phagocytic cup (push and lock). Inset, the pulling forces from the cell (dashed arrow) on the target and paired forces from the target on the cell (solid arrow) likely put the receptor-ligand interactions under tension. After 50% engulfment, the net ring force points inward, and is balanced by compressive forces (cyan arrows) throughout the phagocytic cup.

smaller phagocytic targets, while the uptake of particles comparable in size to those in our study (~10 μm) was Arp2/3-dependent (*Rotty et al., 2017*). This is consistent with our observations that inhibition of the Arp2/3 complex dramatically reduces actin tooth number and size, and ultimately target constriction and uptake efficiency.

Unexpectedly, myosin-II motor inhibition also reduced tooth number and size, albeit to a lesser extent than Arp2/3 inhibition, and also altered tooth spatial distribution (*Figure 4—figure supplement 1*). Inhibition of myosin-II in studies on podosomes also caused a reduction in podosome stiffness and protrusive activity (*Labernadie et al., 2014*; *Labernadie et al., 2010*). In 2D podosomes, myosin-II filaments interconnect radial actin fibers of individual podosomes to create a coherent network (*Labernadie et al., 2014*; *Meddens et al., 2016*). The localization of myosin-II behind and in between teeth, as well as the coordinated movement of teeth at the mesoscale, suggests that actin teeth in phagocytosis may be comparably interconnected by actomyosin structures, bearing similarity to the organization of podosome rosettes (*Linder, 2007*). We find that individual teeth grow larger and stronger with phagocytic progression, correlating with increasing myosin-II constriction observed in late-stage phagocytosis (*Figure 7a*). These observations may relate to the force feedback mechanism observed in Arp2/3-mediated branched actin networks in vitro showing that mechanical resistance makes self-assembling actin networks stronger (*Bieling et al., 2016*; *Li et al., 2021*). Thus,

increased myosin-II contraction in late-stage phagocytosis may promote stronger actin teeth. This rapid structural reinforcement by myosin-II has not yet been described for actin cores of podosomes on 2D, and may be unique to phagocytosis.

Further potential differences between actin teeth and 2D podosomes should also be noted. Although actin teeth are podosome-like in protein composition, the typical ring-like organization of adhesion proteins surrounding the F-actin core of 2D podosomes is not apparent during phagocytosis. Instead, we, and others (*Ostrowski et al., 2019*), observe vinculin, for example, to appear more punctate behind actin teeth during phagocytosis (*Figure 5—figure supplement 1*). This could be due to the difficulty in visualizing the supramolecular architecture in the 3D context of phagocytosis involving curved targets, yet this could also point to differences in the molecular arrangement between actin teeth and 2D podosomes. In the future, higher resolution imaging will be necessary to better compare the architecture of actin teeth and podosomes, including the potential presence of a podosome 'cap' in actin teeth (*Linder and Cervero, 2020*), and may potentially expose changes in mechanosensitive protein localization in response to phagocytic targets of different rigidities. In support of differences between the molecular architecture of actin teeth and podosomes, the nature of force exertion at these structures also likely differs. In 2D podosomes, the protrusive force of the actin 'core' is counterbalanced by pulling forces from the surrounding adhesion complexes (*Bouissou et al., 2017*). This force exertion profile was, at least partly, inferred from the observation of lower global deformation than expected by the protrusive activity of podosome cores alone. In contrast, we observe higher large-scale deformation than expected by the protrusive activity of teeth alone (*Figure 4j*) and the deformations on the DAAMPs are not indicative of strong local tensile forces by adhesion complexes. Finally, we show that actin teeth are mechanosensitive, forming more frequently when cells are challenged with softer targets, contrary to the reported behavior of podosomes in 2D (*Labernadie et al., 2014*; *van den Dries et al., 2019a*; *van den Dries et al., 2019b*; *Figure 4j*).

We further show that myosin-II plays an important role in phagocytic cup closure – a stage in phagocytosis that has been notoriously difficult to study experimentally (*Marie-Anaïs et al., 2016*; *Figure 7a*). Generally, myosin-II is not deemed important for cup closure or phagocytic progression (*Jaumouillé and Waterman, 2020*; *Swanson et al., 1999*), yet we observe myosin-II enrichment at the rim of phagocytic cups (*Figure 5c*). Moreover, Blebbistatin treatment specifically affects late-stage contractile force generation (*Figure 3h*) causing cup closure to become a bottleneck step as evidenced by the accumulation of late-stage phagocytic cups (*Figure 3g*). Because these forces are exerted normal to the target surface, the contribution of myosin-II activity to phagocytic progression has likely gone unnoticed in previous studies using the most common implementation of TFM on a planar substrate, which only measures forces that are tangential to the surface (shear) (*Barger et al., 2019*; *Jaumouillé et al., 2019*; *Kovari et al., 2016*). Seemingly contradictory to its apparent role in force generation at the later stages of cup closure, we do not find that myosin-II inhibition affects uptake efficiency measured as the ratio of fully closed to incompletely closed cups. One potential explanation is that when phagocytosis is stalled beyond 90% engulfment upon myosin-II inhibition, the incomplete closure of the cup may be clearly observed in high-resolution 3D imaging of individual cells, but such late-stage cups may be mistaken for fully internalized particles in the lower resolution 2D microscopy used for evaluating phagocytic efficiency in a large number of cells. Alternatively, a model in which myosin-II has opposing effects on phagocytic progression during early (beneficial) and late-stage (adverse) phagocytosis could also be consistent with our observations. It is also possible that while myosin II-mediated constriction is important for the architecture or normal progression of the phagocytic cup or for cup closure in more challenging environmental contexts, cup closure can still proceed in the absence of functional myosin-II through alternative pathways such as actin polymerization-driven extension of the cup rim. Similarly, in *Dictyostelium* cytokinesis, myosin-II is required for cortical tension generation in the cleavage furrow but cytokinesis can still be successfully completed on adhesive surfaces via alternative mechanisms (*Zang et al., 1997*; *Zhang and Robinson, 2005*). Ultimately, our findings underscore the complexity of the myosin-II contributions to phagocytosis and highlight the power of new biophysical tools to identify its precise functioning.

We show that FcR-mediated phagocytosis is hallmarked by a unique force balance, where before 50% engulfment, target constriction results in an outward force balanced by pulling forces throughout the base of the phagocytic cup (*Figure 7b*). Given the reduced presence of cytoskeletal components at the base of the cup (*Figure 2—figure supplement 1*; *Freeman and Grinstein, 2014*; *Jaumouillé*

*and Waterman, 2020*), these pulling forces are most likely not due to actin polymerization forces or actomyosin contractility, but instead a result of the target being held in place through receptor-ligand bonds throughout the base of the cup. Hence, the forces acting on receptor-ligand bonds within the cup are likely dependent on the target physical properties and the local geometry at the rim of the cup (*Figure 7b*). Since the lifetime of such bonds is tension-dependent (*Nishi et al., 2017*; *Zhu et al., 2019*), this push-and-lock mechanism may enable sensing of physical target properties through a proofreading mechanism and thereby aid macrophages in target selection. Indeed, we observe that forfeit of uptake via popping, in which the outward-directed forces likely overcome the strength of the receptor-ligand interactions, depends on the local target geometry and target rigidity. In addition, for non-spherical targets (e.g. ellipsoidal), this force balance would result in a net torque aligning the target's long axis along the phagocytic axis, which has been observed previously and is reported to lead to enhanced uptake efficiency of ellipsoidal targets (*Champion and Mitragotri, 2006*; *Schuerle et al., 2017*; *Sosale et al., 2015*).

We observed that actin teeth formed more frequently when cells were challenged with softer targets (*Figure 4i*), yet we also associated softer targets with more instances of failed internalization (*Figure 6f*). This calls into question the effectiveness of actin teeth, and, more generally, target constriction, in driving phagocytic internalization. Aside from a role in cup closure, target constriction could be important for creating a tight apposition between the cell and target, which is essential for receptor engagement (*Bakalar et al., 2018*). Surprisingly though, we noticed that overly strong target constriction leads to failure of attempted phagocytosis, expressed either as partial eating or a popping mechanism leading in forfeit of uptake. Although these mechanisms may lead to reduced uptake efficiency in isolated phagocyte-target interaction, we suspect they may be critical in more complex phagocytic encounters that occur in vivo, for example, when multiple macrophages approach a single target (*Figure 6g*), or attempt to engulf adherent and hard-to-reach targets (*Colucci-Guyon et al., 2011*; *Davidson and Wood, 2020*; *Vorselen et al., 2020a*). Altogether, our findings show that actin polymerization-dependent protrusive forces and myosin-II-dependent contractile forces contribute to target deformation and phagocytic internalization, and likely both participate in the mechanosensation required for phagocytic plasticity.

## Code availability

The MATLAB code for analysing confocal images and deriving particle shape is publicly available on https://gitlab.com/dvorselen/DAAMparticle_Shape_Analysis; *Vorselen, 2021*. The Python code used for analysing tractions is provided on https://gitlab.com/micronano_public/ShElastic; *Wang and Cai, 2021*.

# Materials and methods

**Key resources table**

| Reagent type (species) or resource | Designation | Source or reference | Identifiers | Additional information |
|---|---|---|---|---|
| Cell line (*Mus musculus*) | RAW264.7 | ATCC | TIB-71 RRID:CVCL_0493 | Macrophage-like cell line |
| Cell line (*Homo sapiens*) | HL-60 | ATCC | CCL-240 RRID:CVCL_0002 | Differentiated into neutrophil-like cells |
| Biological sample (mouse) | Primary bone marrow cells | *Barger et al., 2019* | | |
| Antibody | Goat polyclonal anti-rabbit-Alexa Fluor-405 | Invitrogen | A31556 RRID:AB_221605 | (1:400) |
| Antibody | Rabbit polyclonal anti-mouse BSA | MP Bio | 0865111 RRID: AB_2335061 | (3 mg/mL) |
| Recombinant DNA reagent | EGFP-myo1e (plasmid) | *Krendel et al., 2007* | | Plasmid construct to transfect and examine myo1e localization |

*Continued on next page*

*Continued*

| Reagent type (species) or resource | Designation | Source or reference | Identifiers | Additional information |
|---|---|---|---|---|
| Recombinant DNA reagent | EGFP-myo1f (plasmid) | *Barger et al., 2019* | | Plasmid construct to transfect and examine myo1f localization |
| Recombinant DNA reagent | mEmerald-Lifeact (plasmid) | Addgene (a gift from Michael Davidson) | #54148 | Plasmid construct to transfect and monitor F-actin dynamics |
| Recombinant DNA reagent | EGFP-RLC (plasmid) | | | Plasmid construct to transfect and examine myosin-II localization |
| Recombinant DNA reagent | pUB-Halo-WASP (plasmid) | This paper | | Plasmid construct to transfect and examine WASP localization |
| Recombinant DNA reagent | CMV-GFP-NMHCII-A (plasmid) | Addgene | #11347 | Plasmid construct to transfect and examine myosin-II localization |
| Recombinant DNA reagent | EGFP-ARP3 (plasmid) | This paper | | Plasmid construct to transfect and examine Arp2/3 complex localization |
| Recombinant DNA reagent | mEmerald-cortactin (plasmid) | This paper | | Plasmid construct to transfect and examine cortactin localization |
| Recombinant DNA reagent | Cofilin-EGFP (plasmid) | Addgene | #50859 | Plasmid construct to transfect and examine cofilin localization |
| Recombinant DNA reagent | mScarlet-i-paxillin (plasmid) | This paper | | Plasmid construct to transfect and examine paxillin localization |
| Recombinant DNA reagent | pUB-mEmerald-vinculin (plasmid) | This paper | | Plasmid construct to transfect and examine vinculin localization |
| Peptide, recombinant protein | BSA | Sigma | A3059 | 20 mg/mL |
| Peptide, recombinant protein | Alexa Fluor-488 conjugated phalloidin | Life Technologies | A12379 | 1:300 |
| Peptide, recombinant protein | Alexa Fluor-568 conjugated phalloidin | Life Technologies | A12380 | 1:300 |
| Peptide, recombinant protein | Alexa Fluor-488 Cadaverine | Thermo Fisher Scientific | A30679 | 0.2 mM final concentration |
| Peptide, recombinant protein | Alexa Fluor-647 Cadaverine | Thermo Fisher Scientific | A30676 | 0.2 mM final concentration |
| Commercial assay or kit | Neon Transfection System 100 µL Kit | Thermo Fisher Scientific | MPK10096 | |
| Chemical compound, drug | CK666 | EMD Millipore | SML0006 | 150 µM |
| Chemical compound, drug | SMIFH2 | EMD Millipore | S4826 | 10 µM |
| Chemical compound, drug | Blebbistatin | EMD Millipore | B0560 | 15 µM |
| Chemical compound, drug | DAAM-particles | *Vorselen et al., 2020b* | | Can be obtained by contacting Julie A Theriot |

*Continued on next page*

*Continued*

| Reagent type (species) or resource | Designation | Source or reference | Identifiers | Additional information |
|---|---|---|---|---|
| Software, algorithm | Fiji (ImageJ) | NIH | RRID:SCR_002285 | |
| Software, algorithm | Imaris | Bitplane | RRID:SCR_007370 | |
| Software, algorithm | MATLAB | Mathworks | RRID:SCR_001622 | |
| Software, algorithm | Python | Python Software foundation | RRID:SCR_008394 | |
| Software, algorithm | Custom MATLAB code | Other | | https://gitlab.com/dvorselen/DAAMparticle_Shape_Analysis |
| Software, algorithm | Custom Python code | Other | | https://gitlab.com/micronano_public/ShElastic |
| Other | VECTASHIELD Antifade Mounting Medium | Vector Laboratories | H-1000 | |

## Cell lines

The RAW 264.7 and HL-60 cell lines were obtained directly from the ATCC and used within passages 3–15. Cell identity was confirmed prior to each experiment using examination of cell morphology by phase-contrast morphology, and cells were regularly tested for mycoplasma contamination using DAPI staining. The cells used in this study do not appear on the list of commonly misidentified cell lines maintained by the International Cell Line Authentication Committee.

## Cell culture

RAW 264.7 (ATCC; male murine cells) were cultured in DMEM, high glucose, containing 10% FBS and 1% antibiotic-antimycotic (Gibco) (cDMEM) at 37°C with 5% $CO_2$. HL-60 cells (ATCC; CCL-240) were cultured in RPMI plus L-glutamine supplemented with 20% FBS and 1% antibiotic-antimycotic (cRPMI). HL-60 cells were differentiated into neutrophil-like cells in culture media containing 1.5% DMSO and used at day 5–6 post-differentiation and plated on 20 μg/mL fibronectin. For the collection of primary murine bone marrow progenitor cells, femurs and tibias of C57BL/6 mice were removed and flushed with cDMEM. Red blood cells were lysed using ACK buffer (0.15 M $NH_4Cl$) and bone marrow progenitor cells were recovered by centrifugation (250× *g*, 5 min, 4°C), washed once with sterile PBS and plated on tissue culture dishes in cDMEM at 37°C with 5% $CO_2$. For differentiation into BMDMs, non-adherent cells were moved to bacteriological Petri dishes the next day and differentiated over 1 week in cDMEM containing 20 ng/mL recombinant murine M-CSF (Biolegend, 576404). Generation of murine BMDCs has been previously described (*Gosavi et al., 2018*). In brief, bone marrow progenitor cells were collected in cRPMI and replated in cRPMI containing 20 ng/mL recombinant murine GM-CSF (Peprotech, 315–03) (DC media). On day 3, DC media was supplemented. On days 6 and 8, half of the culture supernatant and nonadherent cells were spun down and resuspended in cRMPI containing 5 ng/mL GM-CSF. DC maturation was assessed on day 10 by flow cytometry using PE-Cd11c (Biolegend, 117307) and FITC-MHC-II (Biolegend, 107605) staining. Cells were used on day 12. All procedures utilizing mice were performed according to animal protocols approved by the IACUC of SUNY Upstate Medical University and in compliance with all applicable ethical regulations.

## Chemicals and drugs

Blebbistatin, CK666, SMIFH2, and fibronectin were purchased from EMD Millipore. Alexa Fluor-488 and Alexa Fluor-568 conjugated phalloidin were purchased from Life Technologies. Janelia Fluor 549 (JF549) HaloTag Ligand was a generous gift from Luke Lavis.

## Constructs and transfection

Human myo1e and myo1f constructs tagged with EGFP have been previously described (*Barger et al., 2019*). mEmerald-Lifeact was a gift from Michael Davidson (Addgene #54148). Cofilin-EGFP was a gift from James Bamburg (Addgene #50859). Chicken regulatory light chain (RLC) tagged with EGFP was a gift from Klaus Hahn. WASP tagged with myc was a gift from Dianne Cox, and was subcloned into a pUB-Halo-C1 vector. CMV-GFP-NMHCII-A was a gift from Robert Adelstein (Addgene #11347). ARP3-mCherry (Addgene #27682) and mCherry-cortactin (Addgene #27676) were gifts from Christien Merrifield that were subcloned into EGFP-C1 and mEmerald-C1, respectively. Chicken paxillin was a gift from Chris Turner, which was subcloned into mScarlet-i-C1. Chicken vinculin was a gift from Kenneth Yamada (Addgene #50513) and subcloned into pUB-mEmerald-C1. Immunohistochemical staining to determine localization of select podosome-related proteins in relation to the actin teeth did not produce good results, which may be due to the adhesive and porous nature of the IgG-functionalized DAAMPs. As an alternative, we transfected the RAW macrophages with fluorescently tagged proteins. All transfections were accomplished by electroporation (Neon) using the manufacturer's instructions.

## Microparticle synthesis

DAAMPs were synthesized as previously described (*Vorselen et al., 2020b*). First, acrylamide mixtures containing 100 mg/mL acrylic components, 150 mM NaOH, 0.3% (v/v) tetramethylethylenediamine, 150 mM MOPS (prepared from MOPS sodium salt, pH 7.4) were prepared. Mass fraction of acrylic acid was 10% and crosslinker mass fraction was 0.65% or 2.3%, for 1.4 and 6.5 kPa particles, respectively. Prior to extrusion, the gel mixture was degassed for 15 min and then kept under nitrogen atmosphere until the extrusion process was complete. Tubular hydrophobic Shirasu porous glass (SPG) were sonicated under vacuum in *n*-heptane, mounted on an internal pressure micro kit extruder (SPG Technology Co) and immersed into the oil phase (~125 mL) consisting of hexanes (99%) and 3% (v/v) Span 80 (Fluka, 85548 or Sigma-Aldrich, S6760); 10 mL of gel mixture was extruded through SPG membranes under nitrogen pressure of ~7 kPa, 15 kPa, for membranes with pore size 1.9 and 1.4 µm, respectively; 9 µm, 1.4 kPa particles were synthesized using 1.4 µm pore size membranes, whereas 9 µm, 6.5 kPa particles and 11 µm, 1.4 kPa particles were made using 1.9 µm pore size membranes. The oil phase was continuously stirred at 300 rpm and kept under nitrogen atmosphere. After completion of extrusion, the emulsion temperature was increased to 60°C and polymerization was induced by addition of ~225 mg 2,2'-azobisisobutyronitrile (1.5 mg/mL final concentration). The polymerization reaction was continued for 3 hr at 60°C and then at 40°C overnight. Polymerized particles were subsequently washed (5 × in hexanes, 1 × in ethanol), dried under nitrogen flow for ~30 min, and resuspended in PBS (137 mM NaCl, 2.7 mM KCl, 8.0 mM $Na_2HPO_4$, 1.47 mM $KH_2PO_4$, pH 7.4) and stored at 4°C.

## Microparticle functionalization

DAAM-particles were functionalized as previously described (*Vorselen et al., 2020b*). In brief, DAAMPs were diluted to 5% (v/v) concentration and washed twice in activation buffer (100 mM MES, 200 mM NaCl, pH 6.0). They were then incubated for 15 min in activation buffer supplemented with 40 mg/mL 1-ethyl-3-(3-dimethylaminopropyl) carbodiimide, 20 mg/mL *N*-hydroxysuccinimide (NHS), and 0.1% (v/v) Tween 20, while rotating. Afterward they were spun down (16,000 × *g*, 2 min) and washed 4 × in PBS, pH 8 (adjusted with NaOH) with 0.1% Tween 20. Immediately after the final wash, the particles were resuspended in PBS, pH 8 with 20 mg/mL BSA (Sigma, A3059) and incubated, rocking for 1 hr. Then cadaverine conjugate was added: either Alexa Fluor-488 Cadaverine (Thermo Fisher Scientific, A-30679) or Alexa Fluor-647 Cadaverine (Thermo Fisher Scientific, A-30676) to a final concentration of 0.2 mM. After 30 min, unreacted NHS groups were blocked with 100 mM TRIS and 100 mM ethanolamine (pH 9). DAAMPs were then spun down (16,000 × *g*, 2 min) and washed 4 × in PBS, pH 7.4 with 0.1% Tween 20. BSA-functionalized DAAMPs were resuspended in PBS, pH 7.4 without Tween. Finally, DAAMPs were washed 3 × in sterile PBS and opsonized with 3 mg/mL rabbit anti-BSA antibody (MP Biomedicals, 0865111) for 1 hr at room temperature. DAAMPs were then washed 3 × (16,000 × *g*, 2 min) with PBS and resuspended in sterile PBS.

## Phagocytosis assays

DAAMPs were added to a total volume of 400 μL of serum-free DMEM, briefly sonicated in a bath sonicator, and applied to phagocytes in a 12-well plate. To synchronize phagocytosis and initiate DAAMP-phagocyte contact, the plate was spun at 300 × g for 3 min at 4°C. Cells were incubated at 37°C to initiate phagocytosis for a period of 3–5 min. Media was then removed and cells were fixed with 4% PFA/PBS for 15 min. Any unbound DAAMPs were then washed away with 3× washes of PBS and samples were stained with goat anti-rabbit-Alexa Fluor-405 antibodies (Invitrogen, A31556, 1:400) for 30 min to visualize exposed DAAM area. Cells were then washed with PBS (3 × for 5 min each) and permeabilized with 0.1% Triton X-100/PBS for 3 min, then stained with Alexa Fluor-568 or -488 conjugated phalloidin (1:300). Coverslips were then mounted using VECTASHIELD Antifade Mounting Medium (Vector Laboratories, H-1000) and sealed with nail polish. For both live and fixed cell assays, cells were randomly selected for imaging, and all cells were considered, including those having already internalized a particle and those interacting with multiple particles. For drug treatments, cells were exposed to the indicated drug concentration for 30 min prior to the assay and DAAM particles were resuspended and exposed to cells in the same drugged media. For phagocytic efficiency assays, cells were incubated 15, 30, or 60 min before fixation. Similar staining of the exposed DAAM area (using Alexa Fluor-488-AffiniPure Fab Fragment Goat Anti-Rabbit IgG [Jackson Immunoresearch, 111-547-003, 1:1000]) was performed to differentiate fully internalized from adherent and partly internalized particles. Cells were imaged at 20 × magnification and the phagocytic index, defined here as the number of internal particles divided by the number of all cell-associated particles, was determined.

## Microscopy

Confocal images were taken using a PerkinElmer UltraView VoX Spinning Disc Confocal system mounted on a Nikon Eclipse Ti-E microscope equipped with a Hamamatsu C9100-50 EMCCD camera, a 100× (1.4 NA) PlanApo objective, and controlled by Velocity software. Images for protein localization were taken using a Leica TCS SP8 laser scanning confocal microscope with an HC Pl APO 63×/1.4 NA oil CS2 objective at Upstate/Leica Center of Excellence for Advanced Light Microscopy. LLSM images were prone to image artifacts (*Figure 1—figure supplement 1*), and therefore only confocal image data was used for accurate force analysis.

The LLSM (*Chen et al., 2014*) utilized was developed by E Betzig and operated/maintained in the Advanced Imaging Center at the Howard Hughes Medical Institute Janelia Research Campus (Ashburn, VA); 488, 560, or 642 nm diode lasers (MPB Communications) were operated between 40 and 60 mW initial power, with 20–50% acousto-optic tunable filter transmittance. The microscope was equipped with a Special Optics 0.65 NA/3.75 mm water dipping lens, excitation objective, and a Nikon CFI Apo LWD 25 × 1.1 NA water dipping collection objective, which used a 500 mm focal length tube lens. Live cells were imaged in a 37°C-heated, water-coupled bath in FluoroBrite medium (Thermo Scientific) with 0–5% FBS and Pen/Strep. Opsonized DAAMPs were added directly to the media bath prior to acquisition. Images were acquired with a Hamamatsu Orca Flash 4.0 V2 sCMOS cameras in custom-written LabView Software. Post-image deskewing and deconvolution was performed using HHMI Janelia custom software and 10 iterations of the Richardson-Lucy algorithm.

## Microparticle 3D shape reconstruction

Image analysis was performed with custom software in MATLAB, similar to as described previously (*Vorselen et al., 2020b*). Briefly, images were thresholded to estimate the volume and centroid of individual microparticles. Cubic interpolation was then used to calculate the intensity values along lines originating from the particle centroid and crossing the particle edge. Edge coordinates were then directly localized with super-resolution accuracy by fitting a Gaussian to the discrete derivative of these line profiles. This is significantly faster than using pre-processing of the image stacks with the 3D Sobel operator as used previously (*Vorselen et al., 2020b*). Particle properties, such as sphericity, relative elongation and surface curvature, as well as traction forces, are all sensitive to high-frequency noise, so before further calculations edge coordinates were smoothed. To this end, great circle distances ($d$) between edge coordinates with indices $i$ and $j$ were first calculated along a perfect sphere: $d = arccos\left(sin\theta_i sin\theta_j + cos\theta_i cos\theta_j cos\left(\varphi_i - \varphi_j\right)\right)R$, where $R$ is the equivalent radius of a sphere to the particle. Smoothing was then performed by averaging the radial component of the edge coordinates within the given window size (1 μm²), which is similar to a 2D moving average, but

adapted for a spherical surface. A triangulation between edge coordinates was then generated, and the particle surface area $S$ and volume $V$ calculated. Sphericity was calculated as $\Psi = (6\pi^{1/3}V^{2/3}S^{-1})$. For surface curvature calculations, first principal curvatures ($k_1$ and $k_2$) of the triangulated mesh were determined as described previously (*Ben Shabat and Fischer, 2015*; *Rusinkiewicz, 2004*). The mean curvature was calculated $H = (k_1 + k_2)/2$.

## Force analysis

Force calculations were performed using the fast spherical harmonics method within custom Python package ShElastic as described in detail previously (*Vorselen et al., 2020b*; *Wang et al., 2019*). Briefly, to derive both normal and shear forces, we solve the inverse problem of inferring the traction forces ***T*** in an iterative process. We start with a trial displacement field $u$, and during optimization the observed particle shape is always matched exactly, while the following cost function is minimized:

$$f(u) = E_{el} + \alpha R^2 (T; \partial\Omega_t) + \beta E_{pen} (T) \tag{1}$$

where $R(T; \partial\Omega_t)$ are the cellular forces exerted outside of the cell-target contact area, which is obtained from fluorescent actin and immunostaining (*Figure 2—figure supplement 1*), $E_{el}$ is the elastic energy included in this minimization algorithm to penalize unphysical solutions were higher forces producing the same shape, and $\beta E_{pen}$ is an anti-aliasing term. The weighing parameter $\alpha$ for residual traction outside of the cell-target contact area and $\beta$ for anti-aliasing were both 1. Similar to most TFM methods, polyacrylamide gels were assumed to be linearly elastic in the small strain regime ($\varepsilon \lesssim 0.1$) in which force measurements were made. Tractions were calculated with spherical harmonic coefficients up to $l_{max} = 20$, and evaluated on a 21 × 41 grid.

## Fluorescent mapping on particle surface and determination of fraction engulfed

Mapping of fluorescent proteins, phalloidin, and immunostaining to the particle surface was done by determining of the fluorescent intensity along radial lines originating from the particle centroid and passing through each edge coordinate (*Figure 1c*). Linear interpolation was used to determine the intensity along each line, and the maximum value within a 1 µm distance of the edge coordinate was projected onto the surface. The calculation of the fraction engulfed, alignment of particles using the centroid of the contact area, and obtaining of a stress-free boundary for force calculations was done as described previously (*Vorselen et al., 2020b*), with the exception that here both the phalloidin stain and the immunostaining of the free particle surface were used to determine the mask (*Figure 2—figure supplement 1*). For LLSM data, where no staining of the free particle surface was present, alignment was done manually.

## Indentation simulations

Indentation simulations of teeth on a spherical target particle were based on the Hertz contact model (*Johnson, 1985*). Parameters of the model were estimated from experimental data: the undeformed radius of the target particle was set at $R_{target} = 3.7\mu$m; the teeth are considered as rigid spherical indenters with radius $R_{teeth} \approx 0.5\mu$m; and 10 teeth were simulated for each target, which were equally distributed around the equator of the target sphere (*Figure 4—figure supplement 2*). The force $F$ and the contact area radius $a$ produced by indentation to absolute depth $d$ for each individual indenter were then evaluated:

$$F = \tfrac{4}{3}ER^{\frac{1}{2}}d^{\frac{3}{2}}, a = \sqrt[3]{\tfrac{3FR}{4E}},$$

where the effective Young's modulus $E$ and effective radius $R$ are

$$\tfrac{1}{E} = \tfrac{1-\nu_{tooth}^2}{E_{tooth}} + \tfrac{1-\nu_{target}^2}{E_{target}} = \tfrac{3}{4E_{target}}, \tfrac{1}{R} = \tfrac{1}{R_{teeth}} + \tfrac{1}{R_{target}}$$

given that the target particle is near incompressible ($\nu_{target} = 0.5$) and the teeth are rigid compared to the target ($E_{tooth} \gg E_{target}$). Considering non-friction contact, the force distribution on the target sphere in the contact area of each tooth can be written as:

$$p\left(r\right) = p_0 \sqrt{1 - \left(\frac{r}{a}\right)^2}, p_0 = \frac{3F}{2\pi a^2}, r \in \left[0, a\right],$$

where $r$ is the radius to the initial contact point, and $p_0$ is the maximum pressure on the contact plane. Given the resulting traction force map $T\left(\theta, \varphi\right)$ as the boundary condition on the target sphere surface, we solved the elasticity problem, and obtained the displacement map $u\left(\theta, \varphi\right)$ using our ShElastic package (*Figure 4—figure supplement 2*; *Wang et al., 2019*). The $(\theta, \varphi)$ map on the spherical surface has the size of 61 × 121, which is defined by Gauss-Legendre quadrature (*Wieczorek and Meschede, 2018*). Simulations were carried out for a range of tooth radii $R_{teeth}$ and absolute depth $d$ to obtain the effective tooth depth and the average constriction along the equator (*Figure 4—figure supplement 2*), which are directly comparable with experimentally obtained data.

## Acknowledgements

LLSM imaging was performed at the Advanced Imaging Center (AIC) – Howard Hughes Medical Institute (HHMI) Janelia Research Campus. We thank John M Heddleston, Jesse Aaron, and Teng-Leong Chew of the AIC for helpful discussion. The AIC is jointly funded by the Gordon and Betty Moore Foundation and the Howard Hughes Medical Institute. We thank Lorenzo L Labitigan for critical review of the manuscript and Sharon Chase for help with animal experiments.

## Additional information

### Funding

| Funder | Grant reference number | Author |
| --- | --- | --- |
| American Heart Association | Predoctoral fellowship 18PRE34070066 | Sarah R Barger |
| National Institute of Diabetes and Digestive and Kidney Diseases | R01DK083345 | Mira Krendel |
| Associazione Italiana per la Ricerca sul Cancro | Investigator Grant 20716 | Nils C Gauthier |
| Howard Hughes Medical Institute | | Julie A Theriot |
| Cancer Research Institute | CRI Irvington fellowship | Daan Vorselen |
| National Institute of General Medical Sciences | R01GM138652 | Mira Krendel |

The funders had no role in study design, data collection and interpretation, or the decision to submit the work for publication.

### Author contributions

Daan Vorselen, Conceptualization, Formal analysis, Investigation, Methodology, Software, Validation, Visualization, Writing – original draft, Writing – review and editing; Sarah R Barger, Conceptualization, Formal analysis, Investigation, Methodology, Writing – original draft, Writing – review and editing; Yifan Wang, Wei Cai, Formal analysis; Julie A Theriot, Conceptualization, Funding acquisition, Supervision, Writing – original draft, Writing – review and editing; Nils C Gauthier, Conceptualization, Supervision, Writing – original draft, Writing – review and editing; Mira Krendel, Conceptualization, Funding acquisition, Supervision, Writing – original draft, Writing – review and editing, Investigation

### Author ORCIDs

Daan Vorselen http://orcid.org/0000-0002-7800-4023
Sarah R Barger http://orcid.org/0000-0002-6941-7256
Julie A Theriot http://orcid.org/0000-0002-2334-2535
Mira Krendel http://orcid.org/0000-0002-7008-9069

### Ethics

This study was performed in compliance with the recommendations in the Guide for the Care and Use of Laboratory Animals of the National Institutes of Health. All procedures utilizing mice were performed according to the animal protocol (IACUC# 364) approved by the IACUC of SUNY Upstate Medical University and in compliance with all applicable ethical regulations.

### Decision letter and Author response

Decision letter https://doi.org/10.7554/eLife.68627.sa1
Author response https://doi.org/10.7554/eLife.68627.sa2

## Additional files

### Supplementary files
• Transparent reporting form

### Data availability

All quantitative data generated or analysed during this study are included in the manuscript and supporting files. Source Data files contain all numerical data to generate the figures. All confocal image data is available on FigShare repositories (https://doi.org/10.6084/m9.figshare.16666864 and https://doi.org/10.6084/m9.figshare.16677373).

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
