## [Editor Report]

This study provides interesting new insights into the roles of mechanical forces generated by actin polymerization and myosin-II contractility in Fc receptor-mediated phagocytosis. The authors identify F-actin 'teeth', which are important for microparticle constriction throughout the phagocytosis process. Moreover, they elucidate the specific roles of Arp2/3 nucleated actin networks, and myosin-II-based contractile structures in phagocytosis.

---

## [Decision Letter]

**Decision letter after peer review:**

Thank you for submitting your article "Phagocytic 'teeth' and myosin-II 'jaw' power target constriction during phagocytosis" for consideration by *eLife*. Your article has been reviewed by 3 peer reviewers, including Pekka Lappalainen as Reviewing Editor and Reviewer #1, and the evaluation has been overseen by Suzanne Pfeffer as the Senior Editor. The following individual involved in review of your submission has agreed to reveal their identity: Renaud Poincloux (Reviewer #3).

Essential revisions:

1. The data obtained by using pharmacological inhibitors (CK666, SMIFH2 and blebbistatin) are not particularly convincing. The authors should at minimum confirm that the compounds efficiently inhibited Arp2/3 and myosin II with the concentrations used here, as well as carry out phagocytosis tests to see the effects of these compounds for engulfment of particles. Ideally, the phagocytosis experiments would be done by using both soft and rigid particles to gain insight into possible different contributions of the 'teeth' and actomyosin ring in phagocytosis of particles of various stiffness. Finally, the SMIFH2 data should be removed (or moved to supplements), because recent studies demonstrated that this inhibitor is not specific for formins.

2. Because the authors almost exclusively use a single macrophage cell line (RAW cells), some key experiments should be repeated with macrophages derived from blood monocytes.

3. The authors should better discuss their results relative to published literature.

*Reviewer #1 (Recommendations for the authors):*

1. The authors state that the actin-based protrusive 'teeth' are mechanosensitive. However, they apparently only examined the numbers of phagocytic cups with 'teeth' when macrophages were challenged with DAAMPs of 1.4 kPa or 6.5 kPa (Figure 4i). To strengthen this conclusion, a much more thorough comparison of phagocytosis of soft vs. stiff (1.4 kPa vs. 6.5 kPa) microparticles would be informative. The authors should at minimum examine the intensities of F-actin, WASP and Arp2/3 in teeth on soft vs. stiff microparticles, and compare the compressive stresses (assuming that this is feasible also when using 6.5 kPa microparticles).

2. Many conclusions of the study are based on the use of Arp2/3, myosin II, and formin inhibitors. However, from the data presented, it remains unclear how efficient the inhibitor treatments were (i.e. did the inhibitors completely disrupt Arp2/3 and myosin II localizations in the treated cells). Moreover, the authors should be aware that SMIFH2 also inhibits myosins (Nishimura et al., 2021), and this should be taken into account when interpreting the results obtained by using this inhibitor.

3. From the data presented in the manuscript, the roles of Arp2/3 and myosin II for Fc receptor -mediated phagocytosis remain somewhat elusive. Thus, it would be informative to study the effects of Arp2/3 and myosin II inhibitions on the kinetics and efficiency of phagocytosis. Now the authors apparently only analyzed the effects of these inhibitors on the distribution of cup stages.

*Reviewer #2 (Recommendations for the authors):*

1. The first two figures describing the system and presenting the initial findings are outstanding.

2. Although the mechanical role of cargo has been studied a bit by earlier work, the carefulness of the preparation of the particles and their characterization is excellent.

3. I was a little disappointed that they elected to almost exclusively use a single macrophage cell line (RAW cells), when experiments with cells closer to primary macrophages are quite doable and would have been more convincing and relevant to physiological phagocytosis.

4. If the authors were to investigate the role of myo1e and 1F, I think that would improve the paper and make up for some of the deficiencies of the pharmacological treatments.

5. The meal sharing and pooping stuff is cute, but kind of phenomenological and disconnected from the rest of the paper.

As stated above, clean up the pharmacological stuff in Figure 3 and discuss the results relative to published literature in a more complete and nuanced way, I think this is a great paper.

*Reviewer #3 (Recommendations for the authors):*

The description of podosome-like structures in phagocytosis should also be discussed in more detail. Podosome-like structures have already been observed at phagocytic cups (Allen and Aderem A, 1996; Labrousse A. et al., 2011; Ostrowski et al., 2019). Recently they have also been observed on closed phagosomes containing IgG-coated beads but not on phagosomes containing naked beads (Tertrais M et al., 2021). Additionally, actin flashes have been described on of mature phagosomes in RAW 264.7 macrophages ingesting IgG-coated particles (Liebl and Griffiths G, 2009). In contrast, Poirier et al., 2020, proposed that actin flashes form at phagosomes only when phagocytosis is mediated by complement receptors but not by Fc receptors (Poirier et al., 2020). The function of Arp2/3 for the formation of actin flashes and podosome like structures has also been started to be documented in these articles. As the difference in phenotype might arise from differences in cell models (primary macrophages vs macrophage cell lines), I would suggest testing whether human macrophages derived from blood monocytes also form force-generating teeth on both phagocytic cups and closed phagosomes and examine whether they are also Arp2/3 dependent.

The introduction and Discussion sections could also be strengthened by introducing and commenting on the work of Sergio Grinstein (Freeman et al., Cell 2016 and 2018, in addition to Ostrowski et al., Dev Cell 2019, already cited).

Line 52. "Podosomes are… capable of generating traction forces". The work from Labernadie et al., Nat. Commun. 2014, Proag et al., ACS Nano 2015 and Bouissou et al., ACS Nano 2017 argue that podosomes are capable of generating both traction and protrusion forces, and not only traction forces.

Lines 70-71. "TFM only measures forces tangential to the target surface". There are many studies using TFM that also consider the normal forces applied to the surface.

Line 99. "1.4 or 6.5 kPa". Why was this range of stiffness chosen? Is it comparable to the stiffness of more physiological particles (bacteria, apoptotic bodies)?

Line 103. "Interestingly…" This result is presented as a new result but similar observations were made in Vorselen et al., Nat. Commun. 2020. I suggest the authors to clarify the difference with their previous observations.

Line 109. "…400 nm maximum target constriction". This is a large constriction, given the size of the particle. How was this value evaluated? And for such a deformation, is the method of evaluating the normal and shear stresses from the deformation of the particle still applicable? What is the constriction limit above which it is no longer applicable?

Lines 111-112. "0.5 kPa" and "1.5kPa". I suggest mentioning that these are values of shear stresses, to avoid confusion with the Young's moduli of the particles.

Lines 129. "The deformation patterns… were similar to those observed in living cells". I am surprised that chemical fixation and the difference in imaging do not alter the assessment of the forces. I suggest to specify how the resolution (LLSM vs confocal) impacts the force evaluation (near the equatorial plane and at the upper apex), and present side by side the distribution of the deformations observed in live and fixed samples.

Lines 136 and 138. "Compressive forces" and "shear forces". The deduction of forces from the particle deformation involves a model and assumptions. In this case, it is not known precisely where the forces are applied, on which surface, in which direction… I therefore suggest the authors to detail and justify the assumptions that have been chosen for this assessment.

Lines 222-225. Are the forces of similar amplitude in these different cells?

Lines 231-236. The partial reduction of formation of "teeth" with CK666, SMIFH2 and Blebbistatin contrasts with the literature describing a complete inhibition of podosomes with CK666 and very slight effects on podosome formation by SMIFH2 and Blebbistatin. I suggest the author to discuss this interesting discrepancy. Still concerning these treatments, did the authors verify their effect on podosomes? And has their effect on the rate of phagocytosis quantified?

Lines 272-275. "Elasticity theory simulations…" Bouissou et al., argued that there is a local balance of forces at the podosome level (central protrusion and peripheral traction), and not only a protrusion force. Given the similarities of teeth to podosomes, it would be interesting to also consider a similar model of force application.

Lines 299-300. Formins are reputed difficult to localize. Were the authors able to observe expected localizations for these formins, e.g. at filopodia? This lack of localization may be inconsistent with the effect of SMIFH2. Could the authors comment?

Lines 302-304. The localizations of paxillin and vinculin are difficult to interpret and seem different from the vinculin localization observed by Ostrowsky et al., One difference may be the model (mouse macrophages vs human macrophages). Could the authors comment on this and/or try to localize these proteins in the phagocytic cup of human macrophages?

Lines 531-547. Were all cells considered or only those with a single phagocytosed particle?

Line 580. Please specify how much smoothing is done.

Lines 584-585. Why are the coordinates smoothed again? Is it for aesthetic reasons? And what is the impact on the evaluation of forces? Could the authors explain?

Figure 4. The authors claim that actin based teeth are interconnected, but it seems to me that this is not yet proven.

Figure 7. Given the similarities with podosomes, could local traction forces also be considered in this model?

Supp. Figure 1j. Could the authors determine an objective criterion to qualify an image quality appropriate or insufficient?

Supp.Figure 2-5. Would it be possible to average the three types of maps (deformation/actin/normalized stain), after classification, to perhaps allow easy and visual comparison of the effects of the treatments?

Supp. Figure 7-8 "physical properties" is imprecise. What do you mean exactly? F-actin intensity? Size? Please clarify.

---

## [Author Response]

Essential revisions:1. The data obtained by using pharmacological inhibitors (CK666, SMIFH2 and blebbistatin) are not particularly convincing. The authors should at minimum confirm that the compounds efficiently inhibited Arp2/3 and myosin II with the concentrations used here, as well as carry out phagocytosis tests to see the effects of these compounds for engulfment of particles.

We carried out new experiments to investigate the effect of CK666, blebbistatin and SMIFH2 on phagocytic efficiency, which we now present in a new figure panel (figure 3g). The new experiments supplement our findings regarding the precise effects of these inhibitors on the cytoskeletal reorganization and force production during phagocytosis with the additional functional information.

As detailed in our response below, we believe that the extremely strong effects on force production during phagocytosis following the use of CK666 and blebbistatin treatment, described in the previous version of the manuscript, provide clear quantitative evidence of the efficacy of these treatments. We observe almost complete disruption of target deformation upon Arp2/3 treatment, and complete disruption of target constriction upon blebbistatin treatment (Figure 3 b-d,h,i). Moreover, we see additional effects of these drug treatments on actin localization (Figure 3 d-f, Figure 4 d). Overall, we are not concerned that these compounds are failing to inhibit their known targets, because they clearly produce strong, specific, and distinct effects on particle deformations.

Ideally, the phagocytosis experiments would be done by using both soft and rigid particles to gain insight into possible different contributions of the 'teeth' and actomyosin ring in phagocytosis of particles of various stiffness.

As detailed in our response below, we have now performed additional experiments to more thoroughly compare phagocytosis of targets with different rigidity. We have now fully devoted figure 1 – supplement 2 to more detailed analysis of uptake of stiffer targets, and refer to our findings and this figure in various places in the text. There, we directly compared actin intensity and constriction magnitude, and discuss the forces involved in phagocytosis of soft vs. stiff beads (1.4 kPa vs. 6.5 kPa) beads. We further included the effect of drug-treatments on uptake efficiency for both softer and stiffer particles (Figure 3g and figure 1 – supplement 2f).

Finally, the SMIFH2 data should be removed (or moved to supplements), because recent studies demonstrated that this inhibitor is not specific for formins.

Although the very clear differences between direct myosin-II inhibition by blebbistatin and SMIFH2 treatment strongly suggest that our observations using the SMIFH2 treatment are not the result of off-target myosin-II inhibition by SMIFH2, we have followed the reviewer’s suggestion and moved all SMIFH2 inhibition data to the supplemental figures (Figure 3 – supplement 5), and we now mention and cite the recent study that reports off-target effects of SMIFH2.

2. Because the authors almost exclusively use a single macrophage cell line (RAW cells), some key experiments should be repeated with macrophages derived from blood monocytes.

We have performed additional experiments and analyzed additional data to include more details on uptake by primary bone marrow derived macrophages (BMDMs). We have chosen BMDM over human blood monocytes because we do not have the appropriate protocols to conduct experiments on human-derived macrophages, and we have previous experience working with primary murine macrophages derived from bone marrow.

Compared to RAW macrophages, transfecting and imaging the primary cells was non-trivial. Moreover, we observed that force exertion by primary BMDMs is lower than force exertion by RAW macrophages, as evidenced by significantly smaller target deformations. To observe meaningful deformations and derive cellular forces, we had to challenge BMDMs with targets of lower rigidity (0.3 kPa), which are taken up less efficiently (as we and others have reported, phagocytosis is a mechanosensitive process that is less efficient for lower rigidity targets). Nevertheless, we now show that uptake mediated by primary BMDMs largely follows the same mechanical progression, and that they also form teeth on such softer targets (Figure 1 – supplement 3).

In addition to these quantitative differences between RAW and primary macrophages, we also observe some qualitative differences that will require substantial additional work to elucidate (for example BMDMs appear to exert compressive forces at the base of the phagocytic cup, and target constriction doesn’t always appear right at the rim of the cup but can lag significantly behind the cup rim). We now have included a supplementary figure panel to show the typical progression of primary-macrophage mediated phagocytosis (Figure 1 – supplement 3), and briefly mention the similarities and differences in the main text:

“To determine whether this behavior was also present in primary macrophages, we also imaged bone marrow-derived macrophages transfected with mEmerald-Lifeact and observed a similar mechanical progression, albeit with smaller deformations (Figure 1 – supplement 3).”

We believe a more in-depth analysis of primary cells would be best presented in a subsequent manuscript given the already substantial amount of data and new analysis in the current submission.

3. The authors should better discuss their results relative to published literature.

As detailed in our response to specific reviewer comments, we have expanded our introduction and Discussion sections to better discuss our results in light of other published literature, including a more thorough discussion of the similarities and differences between actin teeth and podosomes as well as the role of the Arp2/3 complex and myosin-II in phagocytosis. All changes compared to the original version have been emphasized in blue in the related manuscript file submitted along with the revised manuscript.

Reviewer #1 (Recommendations for the authors):1. The authors state that the actin-based protrusive 'teeth' are mechanosensitive. However, they apparently only examined the numbers of phagocytic cups with 'teeth' when macrophages were challenged with DAAMPs of 1.4 kPa or 6.5 kPa (Figure 4i). To strengthen this conclusion, a much more thorough comparison of phagocytosis of soft vs. stiff (1.4 kPa vs. 6.5 kPa) microparticles would be informative. The authors should at minimum examine the intensities of F-actin, WASP and Arp2/3 in teeth on soft vs. stiff microparticles, and compare the compressive stresses (assuming that this is feasible also when using 6.5 kPa microparticles).

We thank the Reviewer for this helpful suggestion. We have now performed additional experiments to more thoroughly compare phagocytosis of targets varying in rigidity. We have included new figure panels that directly compare actin intensity and constriction magnitude, and discuss the forces involved in phagocytosis of soft vs. stiff beads (1.4 kPa vs. 6.5 kPa) beads. We have now fully devoted Figure 1 – supplement 2 to a more detailed analysis of uptake of stiffer targets, and refer to our findings and this figure in various places in the text. Overall, stiff (6.5 kPa) target uptake shows a qualitatively very similar signature as soft target (1.4 kPa) uptake. The target deformations are (expectedly) smaller for stiffer targets, but we also, interestingly, find less actin accumulation on stiffer targets.

We have also attempted a more systematic analysis of teeth on targets of varying rigidity. This, however, turns out to be a challenging task. We currently automatically detect teeth based on local protrusive activity (as measured by target curvature) and actin intensity (Figure 4 – supplement 1a). However, picking a comparable threshold for identification of teeth on particles of varying rigidities is non-trivial. A key difficulty here is that the stiffer targets deform much less (as now quantified in Figure 1 – supplement 2). This raises the possibility that teeth induce deformations that are below our detection limit, which would not only bias our estimate of tooth numbers, but also the indentation and force generation by the remaining detected teeth. We tested various thresholding strategies for detecting teeth on stiff targets, but this led to different overall conclusions about teeth numbers and protrusive activity. These difficulties in systematic identification and analysis of teeth also make it extremely hard for us to compare WASP and Arp2/3 intensity at teeth as requested by the reviewer. The significant cup-to-cup variation, and difference in transfection efficiency and hence stain intensity between cells, makes such measurements critically dependent on a systematic comparison between a larger number of cups and teeth. We believe that these are hurdles that we may be able to overcome but will require significant further work and therefore in our opinion best reserved for a future study. We have rewritten our paragraph on teeth mechanosensitivity to clarify this point (see pg. 10, line 299).

2. Many conclusions of the study are based on the use of Arp2/3, myosin II, and formin inhibitors. However, from the data presented, it remains unclear how efficient the inhibitor treatments were (i.e. did the inhibitors completely disrupt Arp2/3 and myosin II localizations in the treated cells). Moreover, the authors should be aware that SMIFH2 also inhibits myosins (Nishimura et al., 2021), and this should be taken into account when interpreting the results obtained by using this inhibitor.

As mentioned later in this document, we have decided to move all SMIFH2 data to the SI and note these recent findings regarding SMIFH2 on pg. 8 (line 217). Both CK666 and Blebbistatin inhibit the *activity* of the Arp2/3 complex and myosin II respectively; this mode of action is not expected to ”completely disrupt” target protein *localization*. It is therefore more common to use inhibition of formation of certain structures or force exertion as an indicator of treatment efficacy. For example, the paper in which CK-666 was initially introduced (Nolen et al., 2009) used inhibition of actin structures (podosomes specifically) as a readout.

In our study, we see extremely clear effects on force production after use of CK666 and blebbistatin treatment, which gives a very quantitative readout of the efficacy of these treatments. We observe almost complete disruption of all target deformations upon Arp2/3 treatment, and complete disruption of target constriction upon blebbistatin treatment (Figure 3 b-d,h,i). Moreover, we see additional effects of these drug treatments on actin localization (Figure 3 d-f, Figure 4 d). Altogether, we believe that these data provide very strong evidence of the efficacy of these drugs.

Furthermore, we have now examined the F-actin staining at the ventral portion of the RAW cells in our phagocytosis experiments, and similar to Nolen *et al.,* (2009) we observe complete disassembly of podosome actin structures (marked by red squares) and a dramatic change in cell morphology (increase in spiky filopodia) upon CK666 treatment, as shown in the confocal image slices (Author response image 1).

**Author response image 1. sa2fig1:** 

3. From the data presented in the manuscript, the roles of Arp2/3 and myosin II for Fc receptor -mediated phagocytosis remain somewhat elusive. Thus, it would be informative to study the effects of Arp2/3 and myosin II inhibitions on the kinetics and efficiency of phagocytosis. Now the authors apparently only analyzed the effects of these inhibitors on the distribution of cup stages.

We would like to point out that in addition to the effect of CK666 and Blebbistatin on the distribution of cup stages, in our initial submission we also studied the effect of these inhibitors on (1) actin localization and spatial organization in the phagocytic cup (Figure 3d-f, 4d, Figure 3 – supplement 1b, Figure 4 – supplement 1e), (2) global deformations of phagocytic targets (Figure 4d,i,j Figure 3 – supplement 1b,e), (3) cellular forces generated during phagocytosis (Figure 3a,b), and (4) how all these depend on precise stage of phagocytic progression (Figure 3i,j, Figure 3 – supplement 1c,e, Figure 4 – supplement 1g).

For this revised submission, we have now also included data on the efficiency of phagocytic uptake using the DAAM particles under drug treatments at various timepoints. This data is presented in a new figure panel (3g), Figure 3 – supplement 1d (kinetics), and 2f (stiff particles). We see a very strong inhibition of phagocytic uptake after 15 minutes by the Arp2/3 inhibitor, and no pronounced effect after myosin II inhibition. These observations hold true at later timepoints, and are also very similar when using stiffer 6.5 kPa targets. We now also discuss these results in light of the previous literature in our Discussion section.

Reviewer #2 (Recommendations for the authors):1. The first two figures describing the system and presenting the initial findings are outstanding.2. Although the mechanical role of cargo has been studied a bit by earlier work, the carefulness of the preparation of the particles and their characterization is excellent.

We thank the reviewer for these kind words about our work.

3. I was a little disappointed that they elected to almost exclusively use a single macrophage cell line (RAW cells), when experiments with cells closer to primary macrophages are quite doable and would have been more convincing and relevant to physiological phagocytosis.

We appreciate this criticism and have now included more information on the phagocytic behavior of murine primary bone-marrow derived macrophages (BMDM) (see new Figure 1 – supplement 3). Compared to RAW macrophages, transfecting and imaging the primary cells was non-trivial. Moreover, as evidenced by significantly smaller target deformations, force exertion by primary BMDMs is lower than force exertion by RAW cells. This results in an additional complication, which is that to observe meaningful deformations and derive cellular forces, we had to use targets of lower rigidity (0.3 kPa), which are taken up less efficiently (since as we and others have reported, phagocytosis is a mechanosensitive process that is less efficient for softer targets). Nevertheless, we now show that uptake by primary BMDMs follows a qualitatively similar mechanical progression, including formation of protrusive actin-rich teeth and overall target constriction.

4. If the authors were to investigate the role of myo1e and 1F, I think that would improve the paper and make up for some of the deficiencies of the pharmacological treatments.

Although we agree with the Reviewer that this would be very interesting, we believe that the manuscript, as is, provides sufficient new insights into the biophysics of phagocytosis, as well as the roles of specific actin structures and regulators on phagocytic progression. We worry that adding results on myosin 1e/f would be too ambitious, and feel like this would be more appropriate for a follow-up study.

5. The meal sharing and pooping stuff is cute, but kind of phenomenological and disconnected from the rest of the paper.

We agree with the reviewer that these observations are necessarily less quantitative than most of the analysis in this paper. Since these observations aren’t particularly rare (up to 30% of soft particle uptake), we believe there is value in reporting this behavior. Moreover, the relation between phagocytic force exertion and uptake efficiency is currently poorly understood, and as the reviewer pointed out, the forces generated during phagocytosis by Arp2/3-branched actin polymerization and myosin-II motor proteins could be more important depending on the context in which phagocytosis is taking place. These observations of popping and meal-sharing are an indication of how force exertion may differently affect isolated cell-target interactions than more complex cell-target encounters.

As stated above, clean up the pharmacological stuff in Figure 3 and discuss the results relative to published literature in a more complete and nuanced way, I think this is a great paper.Reviewer #3 (Recommendations for the authors):The description of podosome-like structures in phagocytosis should also be discussed in more detail. Podosome-like structures have already been observed at phagocytic cups (Allen and Aderem A, 1996; Labrousse A. et al., 2011; Ostrowski et al., 2019). Recently they have also been observed on closed phagosomes containing IgG-coated beads but not on phagosomes containing naked beads (Tertrais M et al., 2021). Additionally, actin flashes have been described on of mature phagosomes in RAW 264.7 macrophages ingesting IgG-coated particles (Liebl and Griffiths G, 2009). In contrast, Poirier et al., 2020, proposed that actin flashes form at phagosomes only when phagocytosis is mediated by complement receptors but not by Fc receptors (Poirier et al., 2020). The function of Arp2/3 for the formation of actin flashes and podosome like structures has also been started to be documented in these articles. As the difference in phenotype might arise from differences in cell models (primary macrophages vs macrophage cell lines), I would suggest testing whether human macrophages derived from blood monocytes also form force-generating teeth on both phagocytic cups and closed phagosomes and examine whether they are also Arp2/3 dependent.

We thank the Reviewer for this interesting comment. The frequency or characteristics of actin teeth may well differ among phagocyte cell types, however we would like to point out that we already tested and observed their presence in human neutrophil-like cells and murine primary bone marrow derived macrophages as well as primary dendritic cells during phagocytosis. To address the reviewer’s comments, we have added more data on force generation, evaluated as target deformations, induced by actin teeth in primary murine bone marrow derived macrophages (BMDMs) during live cell lattice light sheet imaging (new figure 1 – supplement 3). Although there are quantitative differences between BMDM and RAW cell force exertion (BMDMs generally induce less target deformation than RAW cells), the qualitative signature, including overall target constriction and observation of distinct actin-rich puncta associated with local protrusive activity, appears very similar. The potential difference between human and murine primary phagocyte behavior is an interesting question, especially given the known differences among these cell types with respect to podosome formation. Because we do not have the approved protocols to conduct experiments with human blood-derived monocytes, potential comparisons will have to be addressed in a future study.

We now have also expanded our discussion of previous observations of podosome-like structures in phagocytosis, including the work by Tertrais et al., (2021). The idea to look at actin structures on closed phagosomes is very interesting. As the reviewer points out, actin flashes have been observed around phagocytic targets coated with different ligands, including bacteria like *Listeria*. These actin flashes may also be Arp2/3-mediated, and may, as the work by Tetrais et al., reports, contain podosome like structures. In addition, we have previously seen distinct deformations of targets after completion of engulfment (Vorselen et al., 2020). These deformations far exceeded the deformations caused by phagocytic teeth and originated from the site of cup closure. Together, to our knowledge there are still many unknowns around the regulation and dynamics of actin structures on closed phagosomes, and how this depends on target properties and engulfment mechanisms. Albeit very interesting, going into the actin dynamics associated with relocation and maturation of phagosomes and the potential role of teeth or podosome-like structures in that process, would, in our opinion, warrant an entire study by itself.

The introduction and Discussion sections could also be strengthened by introducing and commenting on the work of Sergio Grinstein (Freeman et al., Cell 2016 and 2018, in addition to Ostrowski et al., Dev Cell 2019, already cited).

We have now rewritten the introduction and Discussion sections to more fully comment on this past literature, including the suggested works. Thank you for this suggestion.

Line 52. "Podosomes are… capable of generating traction forces". The work from Labernadie et al., Nat. Commun. 2014, Proag et al., ACS Nano 2015 and Bouissou et al., ACS Nano 2017 argue that podosomes are capable of generating both traction and protrusion forces, and not only traction forces.

We have removed the word traction from this sentence. We now also discuss the similarities and potential differences between actin tooth and podosome force exertion in our Discussion section.

Lines 70-71. "TFM only measures forces tangential to the target surface". There are many studies using TFM that also consider the normal forces applied to the surface.

We thank the reviewer for pointing out this mistake. We fully agree and are aware of applications of TFM that do measure normal forces. We have adapted this sentence to reflect this: “Moreover, in its most common application, TFM only measures forces tangential to the target surface, neglecting forces normal to the target surface, which may be critical.”

Line 99. "1.4 or 6.5 kPa". Why was this range of stiffness chosen? Is it comparable to the stiffness of more physiological particles (bacteria, apoptotic bodies)?

The stiffness range chosen is similar to healthy tissue cells as well as apoptotic cells. We added the following to clarify this:

“RAW macrophages were fed DAAMPs with a diameter of 9 µm and a Young’s modulus of 1.4 or 6.5 kPa, which is in the same range as tissue cells and physiological targets of macrophages like apoptotic cells^34,35^. Raw macrophages typically formed a chalice-shaped phagocytic cup and completed phagocytosis in similar timeframe (~3 minutes) as previously reported for much stiffer (2-3 GPa) polystyrene particles^36^ (Figure 1a, Video 1, Figure 1 – supplement 1e).”

In addition, there are also practical reasons to use relatively soft particles because deformations on much stiffer particles are not well observed and hence forces cannot be inferred.

Line 103. "Interestingly…" This result is presented as a new result but similar observations were made in Vorselen et al., Nat. Commun. 2020. I suggest the authors to clarify the difference with their previous observations.

Our previous work was mostly focused on establishing the microparticle traction force technology. Indeed, a similar observation was made as those described here, but the previous measurements were of a single phagocytic cup on a fixed cell. While we believe that this current description as a common feature of phagocytosis is still novel, we have moved the word “Interestingly” to the more novel discovery of the teeth movement by live-cell imaging. We have also included the following note in this section: “While these deformations were more apparent using the softer 1.4 kPa DAAMPs, the same force pattern was observed using stiffer 6.5 kPa targets (Figure 1 – supplement 2a-b, Video 3) and previously using 0.3 kPa targets using J774 macrophages.”

Line 109. "…400 nm maximum target constriction". This is a large constriction, given the size of the particle. How was this value evaluated? And for such a deformation, is the method of evaluating the normal and shear stresses from the deformation of the particle still applicable? What is the constriction limit above which it is no longer applicable?

We thank the reviewer for this comment and are aware of the limitations of our methodology, which assumes linear elastic particle behavior, for very large deformations, where this assumption is likely to fail.

Firstly, we added a clarification to figure 1 – supplement 1c, which contains the data on maximum target constriction, as to how this value was determined:

“. For each video, the single timepoint in which target constriction was maximal was identified, and the exact constriction value was determined as described in Figure 3 – supplement 1a.”

A typical rule of thumb is that linear elasticity can be assumed for strains of up to 0.1. In our case, a maximum constriction of 400 nm, would amount to a strain of 0.09: still within this range. In addition, this is the constriction in a single frame with maximal constriction selected from a video, and the vast majority of images captured in fixed cell data (which were exclusively used for force calculations) will hence have considerably lower deformation. The constriction values observed in fixed cell data are presented in Figure 2i, and typically ~200 nm (strains < 0.05), which is well within the limits were particle behavior can be reasonably assumed to be linearly elastic. We added a clarification to our methods section regarding this assumption:

“Similar to most TFM methods, polyacrylamide gels were assumed to be linearly elastic in the small strain regime (ε ≲ 0.1) in which force measurements were made.”

Lines 111-112. "0.5 kPa" and "1.5kPa". I suggest mentioning that these are values of shear stresses, to avoid confusion with the Young's moduli of the particles.

The stresses mentioned on line 111 and 112 (now line 142 and 143) are not shear stresses but bulk compressive stresses. Our methodology allows tracking the overall change of particle volume and combined with measuring the bulk modulus of the particles (which was done previously, Vorselen et al., 2020) allows determining bulk compressive stresses. To clarify this point, we changed the wording of this sentence: “In addition, we observed bulk compressive stresses during phagocytic cup progression (~ 0.5 kPa) and after complete internalization of the DAAMPs (up to 1.5 kPa), leading to a dramatic reduction in DAAMP diameter (Figure 1 – supplement 1e-f).”

Lines 129. "The deformation patterns… were similar to those observed in living cells". I am surprised that chemical fixation and the difference in imaging do not alter the assessment of the forces. I suggest to specify how the resolution (LLSM vs confocal) impacts the force evaluation (near the equatorial plane and at the upper apex), and present side by side the distribution of the deformations observed in live and fixed samples.

We thank the reviewer for this comment. We would like to note that the difference in particle reconstruction, and later force evaluation, is not only dependent on resolution, and instead relies on many factors, including signal-to-noise ratio (as detailed in Vorselen et al., 2020) and potential imaging artifacts, as observed in the majority of the lattice light sheet data presented here. Specifically imaging artifacts, which can result in large apparent target deformations outside the cell-target contact area, can make it impossible to converge to a solution when solving the elasticity theory problem, which is the reason we did not include any force calculations based on the LLSM data.

As illustrated in figure 2 – supplement 1b, there is also significant particle-to-particle variation, so a comparison between the live cell data and fixed cell data needs to be based on a systematic analysis. This is why we choose, next to qualitative comparison of the deformation patterns, to quantitatively compare target constriction at ~50% engulfment in both live and fixed cells (Figure 1 – supplement 1D). This is a relatively straight-forward deformation to measure quantitatively, and revealed 250 nm average target constriction in the live-cell data and a ~40% lower 180 nm average constriction in the fixed-cell data. We have updated the main text, and the supplemental figure to make this comparison clearer:

“The deformation patterns, although slightly lower in magnitude for the fixed samples (Figure 1 – supplement 1), were similar to those observed in living cells by the LLSM imaging (Figure 2b, Figure 1 – supplement 1).”

“d**,** Average target constriction at 50% engulfment. Average is 0.25 ± 0.04 μm (s.e.m., n = 23 cups), which is ~40% higher than observed with fixed cell data at similar stage at 0.18 ± 0.02 μm (s.e.m., n = 19) (main text figure 2i). This indicates partial relaxation of the particle to a more spherical shape during fixation, and suggests that the fixed cell force measurements are a slight underestimate of the real phagocytic forces.”

We also added in the caption of figure 1 – supplement 1h:

“Shape reconstruction in MP-TFM is critically dependent on particle edge localization detection, which shows clear irregularities (yellow arrows) because of the striping artifact. Such artifacts result in large apparent particle deformations inside and outside the cell-target contact area, and can make it impossible to converge on a solution when solving the elasticity theory problem to infer tractions from target deformations. Because of this, we only used the confocal image data for force analysis.”

Lines 136 and 138. "Compressive forces" and "shear forces". The deduction of forces from the particle deformation involves a model and assumptions. In this case, it is not known precisely where the forces are applied, on which surface, in which direction… I therefore suggest the authors to detail and justify the assumptions that have been chosen for this assessment.

To make our methodology and assumptions clearer, we have added a section in the methods (Force analysis) to give more detail on the methodology and assumptions that go into it. We also added a note on previous line 136, now line 164, (“*see methods*”). We do not make any explicit assumptions about the direction and location where the cell applies forces, other than that we restrict them to the cell-target contact area, which is determined from F-actin and exposed particle immunostaining (Figure 2 – supplement 1).

“Force calculations were performed using the spherical harmonics method within custom Python package ShElastic as described in detail previously^21,66^. Briefly, to derive forces, we solve the inverse problem of inferring the tractions in an iterative process to closely match the observed particle shapes and by limiting cellular force exertion to the cell-target contact area as obtained from fluorescent phalloidin and exposed particle immunostaining. In addition, the elastic energy is included in this minimization algorithm to penalize unphysical solutions where higher forces produce the same shape. Similar to most TFM methods, polyacrylamide gels were assumed to be linearly elastic in the small strain regime (ε ≲ 0.1) in which force measurements were made. Tractions were calculated with spherical harmonic coefficients up to l_max_ = 20, and evaluated on a 21 × 41 grid. Chosen values for the weighing parameter a for residual traction outside of the cell-target contact area and b for anti-aliasing were both 1.”

For a more in-depth discussion of the methodology, we also refer to our previous work (Vorselen et al., 2020), in which the methodology is described in much more detail and its performance evaluated using test cases.

Lines 222-225. Are the forces of similar amplitude in these different cells?

The force analysis suggested by the reviewer would be very interesting. However, such an analysis would require larger data sets and careful analysis. As observed in the RAW cell data, we see significant cell-to-cell variation, and dependence on other factors, such as cup stage, of force exertion. Moreover, we currently do not know if there are differences in the mechanical progression of phagocytosis depending on the cell type, beyond tooth involvement. Given that such differences are likely to occur, and given that it is very challenging to decouple force exertion by various cytoskeletal structures (teeth and the contractile ring, for example), it is non-trivial to make direct comparisons between force exertion by teeth between these cell lines. We believe that a comparison of force exertion between different cell types requires an in-depth biophysical comparison between phagocytosis by these cell types, and is beyond the scope of the current manuscript, but would be extremely interesting for future studies.

Lines 231-236. The partial reduction of formation of "teeth" with CK666, SMIFH2 and Blebbistatin contrasts with the literature describing a complete inhibition of podosomes with CK666 and very slight effects on podosome formation by SMIFH2 and Blebbistatin. I suggest the author to discuss this interesting discrepancy. Still concerning these treatments, did the authors verify their effect on podosomes? And has their effect on the rate of phagocytosis quantified?

We have moved all SMIFH2 data to supplemental figures, given the recent report of off-target effects of SMIFH2 treatment, so in our response to this comment we focus on the effect of CK666 and Blebbistatin treatment. Firstly, we note that the reduction of tooth numbers in our experiments is larger for CK666 treatment than Blebbistatin treatment, with the number of teeth is close to 0 for CK666 treatment. Of note, the concentration of CK-666 used in this study inhibits all podosome-like actin structures on the ventral side of RAW macrophages (see earlier Figure).

Differences in findings between our tooth quantification and previous podosome quantification are likely at least partially based on the method of identification of teeth/podosomes. We use a combined metric based on both protrusive activity and F-actin intensity. This is necessary, because at the location of the teeth (the front of the phagocytic cup) actin itself is already strongly enriched. Given these different ways of identification of actin structures, but also potential other differences between these experiments, we are hesitant to make direct quantitative comparisons between these studies.

Nevertheless, we now address the effect of Blebbistatin treatment on podosomes and teeth in our Discussion section. Given that we use protrusive activity in tooth identification, we believe our results are consistent with previous reports on podosome activity, which we now note in the manuscript:

“Unexpectedly, myosin-II motor inhibition also reduces tooth number and size, albeit to a lesser extent than Arp2/3 inhibition, and also alters tooth spatial distribution (Figure 4 – supplement 1). Similarly, inhibition of myosin-II in studies on podosomes cause a reduction in podosome stiffness and protrusive activity^18,58^.”

In addition, actin teeth and podosomes may have significant differences, which we now also discuss more thoroughly in our Discussion sections paragraph starting at line 441, (page 15). Finally, we have performed additional experiments to investigate the effect of these inhibitors on phagocytic efficiency, and we present this data in new figure panels 3g and Figure 3 – supplement 1d. In these experiments we find that the CK-666 treatment has a much more dramatic effect on the phagocytic uptake than the Blebbistatin or SMIFH2 treatments.

Lines 272-275. "Elasticity theory simulations…" Bouissou et al., argued that there is a local balance of forces at the podosome level (central protrusion and peripheral traction), and not only a protrusion force. Given the similarities of teeth to podosomes, it would be interesting to also consider a similar model of force application.

We thank the Reviewer for bringing up this very interesting point. In the work by Bouissou *et al.*, the argument for the local balance of forces at the podosome level is, at least in part, based on the observation that local bulges around each podosome, instead of a large-scale bulge, is only expected if the protrusive forces are locally balanced around each podosome. We believe that this points to an interesting difference with our observations in phagocytosis. As we show in Figure 4j, we see higher, not less, large-scale deformation than expected by the protrusive activity of teeth alone. This suggests that there may be significant differences in the force balance between podosomes and phagocytic teeth. We now discuss this interesting difference and its implications in the Discussion section (lines 454-459). We also expanded our discussion of the more general similarities and differences between podosomes and phagocytic teeth.

Lines 299-300. Formins are reputed difficult to localize. Were the authors able to observe expected localizations for these formins, e.g. at filopodia? This lack of localization may be inconsistent with the effect of SMIFH2. Could the authors comment?

We agree with the Reviewer on this point of inconsistency. We have no reason to believe that the formin constructs we used in this manuscript were unfit for localization studies. To note, mDia1, but not mDia2, has been observed to localize to the phagocytic cups during complement-mediated phagocytosis (Jaumouille et al., 2019). mDia2 and FHOD1 have been shown not to localize to the actin cores of macrophage podosomes. We believe formins such as INF2 or FMNL1, which have been described in macrophage podosomes, are likely to localize to the actin teeth and thus may play a role in actin nucleation. However, to avoid any confusion on this topic, we have removed the mDia2 and FHOD1 localization figures and moved all SMIFH2 data to the SI (Figure 3 – supplement 5).

Lines 302-304. The localizations of paxillin and vinculin are difficult to interpret and seem different from the vinculin localization observed by Ostrowsky et al., One difference may be the model (mouse macrophages vs human macrophages). Could the authors comment on this and/or try to localize these proteins in the phagocytic cup of human macrophages?

There may be many reasons for differences in localization observed between our experiments and those by Ostrowski *et al.,* including cell type (mouse vs. human), but also target rigidity (1.3 kPa DAAMP vs. > 1 GPa polystyrene bead), as well as antibody choice and imaging modality (confocal vs. STED). Moreover, we note that the majority of the work done in the manuscript by Ostrowski was in frustrated phagocytosis assays, and the ring-like distribution of adhesion proteins around an actin-core typically associated with podosomes was much less clear in their assays with spherical targets, which could be due to either technical differences in imaging, or that the structures that are formed during frustrated phagocytosis assays are not identical to those formed during phagocytosis of a target with more physiological curvature. We believe that the results of their localization experiments on spherical targets are more in line with our results. To address this Reviewer’s concern, we have replaced the confocal image of vinculin recruitment in the SI with one that is hopefully clearer in illustrating this point (Figure 5 – Supplement 1c), and now comment on the comparison of the protein distributions we observe with those reported for podosomes and in phagocytosis previously in the Discussion section (lines 441-452).

Lines 531-547. Were all cells considered or only those with a single phagocytosed particle?

To address this point, we added the following brief clarification in our “Phagocytic assays” section of the methods:

“For both live and fixed cell assays, cells were randomly selected for imaging, and all cells were considered, including those having already internalized a particle and those interacting with multiple particles.”

Line 580. Please specify how much smoothing is done.

Please see our reply on line 584-585 below.

Lines 584-585. Why are the coordinates smoothed again? Is it for aesthetic reasons? And what is the impact on the evaluation of forces? Could the authors explain?

We thank the reviewer for their careful attention to the methods section. We now realize this section was unclear and have adapted the wording to clarify the following: only a single smoothing step of the edge coordinates was performed, and original line 580 and 584-584 were referring to a single procedure, where lines 584-585 were detailing the method of smoothing and amount of smoothing performed. We rewrote this section to make this clearer:

“Particle properties, such as sphericity, relative elongation and surface curvature, as well as traction forces, are all sensitive to high-frequency noise, so before further calculations edge coordinates were smoothed. To this end, great circle distances (d) between edge coordinates with indices i and j were first calculated along a perfect sphere: d= arccos⁡(sin⁡θisin⁡θj+ cos⁡θicos⁡θjcos⁡(φi−φj))R, where R is the equivalent radius of a sphere to the particle. Smoothing was then performed by averaging the radial component of the edge coordinates within the given window size (1 µm^2^), which is similar to a 2D moving average, but adapted for a spherical surface.”

More generally, calculation of sphericity, relative elongation, surface curvature and traction forces are all sensitive to high frequency noise. For the latter two, this is largely due to the fact that the equivalent of a derivative is used in these calculations, where taking a discrete derivative of noisy data is bound to amplifying the noise. In addition, for computational efficiency traction force calculations are done using spherical harmonic coefficients up to *l*_max_ = 20 on a 21 x 41 grid. This uses only ~900 points, which is fewer than the ~5000 vertices determined in the original particle shape, and smoothing helps to better reflect the somewhat larger scale deformations of the particle surface.

Smoothing also helps for accurate calculation of relative elongation. This metric (determined by taking the ratio of the length of the phagocytic axis to the average of the two orthogonal axes), is prone to individual outlying vertices. Sphericity is also highly affected by high frequency contributions as those strongly affect the surface area calculation but have minimal effect on the volume.

We believe that an in depth discussion of these factors goes beyond the scope of this paper, but we have added a short clarification (see the section above). In addition, we added the clarification that force calculations were done up using spherical harmonic coefficients up to *l*_max_ = 20 on a 21 x 41 grid.

Figure 4. The authors claim that actin based teeth are interconnected, but it seems to me that this is not yet proven.

We agree with the Reviewer that we have not provided definitive evidence to conclude that teeth are interconnected. Therefore, we have now adapted the phrasing in Figure 4, the abstract, and our model to reflect that teeth are “likely interconnected”. However, we believe that our data tracking the teeth, and specifically that teeth within the same phagocytic cup appeared to move in a coordinated fashion, with similar speed and direction, and even with observed collective speed changes, supports a model where the teeth are functionally interconnected.

Figure 7. Given the similarities with podosomes, could local traction forces also be considered in this model?

As alluded to in our response to the reviewer comment regarding line 272-275, we believe that there currently is not enough evidence to include local tensile forces in this model.

Supp. Figure 1j. Could the authors determine an objective criterion to qualify an image quality appropriate or insufficient?

We thank the reviewer for this suggestion. In this case, because of the image artifacts present in the LLSM data were frequently observed, we were not able to use any of the lattice light sheet videos for force analysis. We therefore believe that our current criterium: to only use the confocal imaging data for force analysis, is a clear and justifiable choice. Coming up with an objective criterion is not trivial in the context of these artifacts that seem to be unique to lattice light sheet microscopy and their origin is not well understood. We changed the wording in the figure panel 1j, and its caption, to clarify this. We also removed the upper row of the figure panels to clarify the figure and message (Figure 1 – supplement 1e).

Supp.Figure 2-5. Would it be possible to average the three types of maps (deformation/actin/normalized stain), after classification, to perhaps allow easy and visual comparison of the effects of the treatments?

We appreciate this suggestion and the value of trying to capture the signature of these complex maps in a simplified manner. Since the particles are only aligned along the phagocytic axis, particles have an arbitrary position with respect to rotation around this axis. Therefore, any features orthogonal to the phagocytic axis will be mostly averaged out when averaging the maps. In addition, since each of these cups is in a different stage of engulfment (ranging from <20% to >90%), averaging the maps directly, without some normalization for the fraction engulfment, is likely not very informative. We believe that visualization of the deformation, curvature and fluorescent intensity profiles along the phagocytic axis (averaged in the directions orthogonal to the phagocytic axis) and then normalized to the fraction engulfed, is the most natural way for easy and visual comparison of the treatments, and this is how we choose to present this data in figure 2f and figure 3d in the main text.

Supp. Figure 7-8 "physical properties" is imprecise. What do you mean exactly? F-actin intensity? Size? Please clarify.

To address the reviewers concern we have changed this wording to “teeth number, size and indentation”.